# Siglec-6 mediates the uptake of extracellular vesicles through a noncanonical glycolipid binding pocket

Edward N. Schmidt[1], Dimitra Lamprinaki[1], Kelli A. McCord [1], Maju Joe [1], Mirat Sojitra [1], Ayk Waldow[1], Jasmine Nguyen [2], John Monyror [3,4], Elena N. Kitova [1], Fahima Mozaneh[1], Xue Yan Guo[1], Jaesoo Jung[1], Jhon R. Enterina[5], Gour C. Daskhan[1], Ling Han[1], Amanda R. Krysler [3], Christopher R. Cromwell [3], Basil P. Hubbard [3,11], Lori J. West [5,6,7], Marianne Kulka [5,8], Simonetta Sipione[3,4], John S. Klassen[1], Ratmir Derda [1], Todd L. Lowary [1,9,10], Lara K. Mahal [1], Meghan R. Riddell [2] & Matthew S. Macauley [1,4,5] ✉

Immunomodulatory Siglecs are controlled by their glycoprotein and glycolipid ligands. Siglec-glycolipid interactions are often studied outside the context of a lipid bilayer, missing the complex behaviors of glycolipids in a membrane. Through optimizing a liposomal formulation to dissect Siglec–glycolipid interactions, it is shown that Siglec-6 can recognize glycolipids independent of its canonical binding pocket, suggesting that Siglec-6 possesses a secondary binding pocket tailored for recognizing glycolipids in a bilayer. A panel of synthetic neoglycolipids is used to probe the specificity of this glycolipid binding pocket on Siglec-6, leading to the development of a neoglycolipid with higher avidity for Siglec-6 compared to natural glycolipids. This neoglycolipid facilitates the delivery of liposomes to Siglec-6 on human mast cells, memory B-cells and placental syncytiotrophoblasts. A physiological relevance for glycolipid recognition by Siglec-6 is revealed for the binding and internalization of extracellular vesicles. These results demonstrate a unique and physiologically relevant ability of Siglec-6 to recognize glycolipids in a membrane.

Sialic acid-binding immunoglobulin-type lectins (Siglecs) are a family of immuno-modulatory cell surface receptors that recognize sialylated glycan ligands[1]. A leading hypothesis for the role of Siglecs in maintaining immune homeostasis is that binding to sialylated glycans represents a form of 'self' recognition that operates as an immune checkpoint[2]. Sialic acid residues decorate proteins and lipids, both of which can act as Siglec ligands[3]. In all cases, sialoside recognition by Siglecs is critically dependent on a conserved canonical arginine residue in their N-terminal V-set domain that forms an ionic interaction between the cationic guanidinium of the essential arginine in Siglecs

[1]Department of Chemistry, University of Alberta, Edmonton, AB, Canada. [2]Department of Obstetrics & Gynaecology and Physiology University of Alberta, Edmonton, AB, Canada. [3]Department of Pharmacology, University of Alberta, Edmonton, AB, Canada. [4]Neuroscience and Mental Health Institute, University of Alberta, Edmonton, AB, Canada. [5]Department of Medical Microbiology and Immunology, University of Alberta, Edmonton, AB, Canada. [6]Department of Pediatrics, University of Alberta, Edmonton, AB, Canada. [7]Department of Surgery, University of Alberta, Edmonton, AB, Canada. [8]National Research Council, Edmonton, AB, Canada. [9]Institute of Biological Chemistry, Academia Sinica, Nangang, Taipei, Taiwan. [10]Institute of Biochemical Sciences, National Taiwan University, Taipei, Taiwan. [11]Present address: Department of Pharmacology and Toxicology, University of Toronto, Toronto, ON, Canada. ✉e-mail: macauley@ualberta.ca

and the anionic carboxylate of the sialic acid[4]. In addition to their physiological roles in human health, Siglecs also play key roles in pathophysiological conditions, as they can be exploited by viruses[5–7], bacteria[8], and cancers[2] for immune evasion. Despite the growing understanding of the roles of Siglecs, there remains an incomplete description of their glycan ligands.

Due to the relatively weak affinity of Siglecs for their glycan ligands, Siglecs are often studied outside of their natural context using approaches that leverage multivalency. This is particularly true for glycolipids as there are challenges associated with studying the binding of Siglecs to glycolipids in a lipid bilayer[9]. Accordingly, the majority of Siglec−glycolipid interactions have been established through plate-based approaches in which a soluble, recombinant Siglec is used to probe glycolipids or neoglycolipids adsorbed on a hydrophobic surface[10–16] or via glycan microarrays where the oligosaccharide portion of the glycolipid is covalently linked to a surface[17]. Using these approaches, Siglec-1, -4, -5, -7, -9, -10, and -15 have been reported to bind the oligosaccharide portion of gangliosides[10,12,18,19], the major class of sialylated glycolipids in mammals. Gangliosides are defined by a core carbohydrate backbone consisting of β-Gal$p$-(1 → 3)-β-Gal$p$NAc-(1 → 4)-β-Gal$p$-(1 → 4)-β-Glc$p$ linked to ceramide[20]. However, only some Siglec−ganglioside interactions have been validated in the context of a lipid bilayer, and even fewer have been studied in a biological membrane. Beyond the challenges associated with studying the two species in a lipid bilayer, deconvoluting Siglec−ganglioside interactions in a biological setting is further complicated by the cell-type-specific combination of gangliosides[21] and expression of multiple Siglecs on immune cells[2].

Despite the challenges associated with studying Siglec−ganglioside interactions, several biological roles have been credited to them. For example, Siglec-1 (CD169/Sialoadhesin; Sn) on macrophages/dendritic cells mediates the uptake of viruses and extracellular vesicles (EVs) through binding gangliosides in their membrane[5,6,22–24]. Siglec-4 (myelin-associated glycoprotein; MAG) on oligodendrocytes binds gangliosides on neurons to regulate neurite growth[25]. Moreover, Siglec-7 on natural killer cells recognizes gangliosides on cancer cells or EVs from cancer cells to prevent immune cell activation[24,26]. The fact that gangliosides are abundant in all mammalian cells makes understanding Siglec−ganglioside interactions important due to their potential of serving broader immunomodulatory roles.

Liposomal nanoparticles have recently emerged as a tool for studying Siglec−ganglioside interactions in a more biological context but have had limited use in studying interactions with Siglec-1[27,28]. Given that presentation of gangliosides from a lipid bilayer can be influenced by the composition of the membrane[29], we hypothesized that systematic optimization of the liposomal formulation may help to reveal a more complete description of Siglec−ganglioside interactions. Here, an optimized liposomal formulation is developed and enables the profiling of all human Siglecs against a panel of gangliosides. Notably, Siglec-6 (CD327, OB-BP1) is shown to bind glycolipids in a noncanonical manner. A series of non-natural, synthetic glycolipids−neoglycolipids−enables the specificity of this glycolipid-binding pocket on Siglec-6 to be probed, culminating in the development of a neoglycolipid ligand for targeting liposomes to Siglec-6-expressing human cells and tissues. We also show that the noncanonical glycolipid binding pocket on Siglec-6 mediates the internalization of EVs, providing a means by which Siglec-6 can participate in immunological tolerance.

## Results

### Optimizing ganglioside liposome binding to Siglec-expressing cells

A panel of 24 Chinese Hamster Ovary (CHO) cells was developed where each cell line expresses a full-length, membrane-bound wildtype (WT)

human Siglec or their corresponding arginine mutant, wherein the canonical arginine residue critical for sialic acid recognition is mutated[30] (Supplementary Fig. 1). CHO cells were chosen as they lack Siglecs and are easy to stably transfect[31]. The extracellular domains of Siglec-14 and -16 are nearly identical to Siglec-5 and -11, respectively; therefore, Siglec-14 and -16 were not included in our panel. Siglec-15 requires the adapter protein DAP12 for cell surface expression due to lysine in its transmembrane segment[32], but this DAP12-dependency was eliminated through a K274L mutation.

An FDA-approved liposomal formulation served as the starting point for our glycolipid-containing liposomes (GLLs), which consisted of 57 mol% 1,2-distearoyl-sn-glycero-3-phosphocholine (DSPC), 38 mol% cholesterol, and 5 mol% polyethylene glycol distearoylphosphatidylethanolamine (PEG$_{45}$−DSPE MW 2000)[33]. This formulation has been reliably used to target Siglecs with high-affinity glycan ligands linked to PEGylated lipids[34,35]. To detect liposome binding to cells by flow cytometry in a cell assay, 0.1 mol% AF647−PEG$_{45}$−DSPE (1) was included in the formulation (Fig. 1a).

As Siglec-1 is an established ganglioside binder[10,36,37], it was used to optimize the liposomal formulation. Liposomes displaying a previously developed high-affinity Siglec-1 ligand[38] linked to PEG$_{45}$−DSPE (2) showed robust binding to WT Siglec-1 and no binding when the canonical essential arginine (Arg116) was mutated to alanine (Supplementary Fig. 2). When (2) was replaced in this formulation with GM1 (3 mol%), a known ligand for Siglec-1[27], little to no binding of these GLLs was observed to WT Siglec-1 CHO cells (Fig. 1b). To test if the hydrodynamic shell created by the PEG$_{45}$−DSPE[39] prevents binding, the amount of PEG in the GLL formulation was systematically decreased, revealing GLL binding. No further increase in GLL binding was observed when PEG$_{45}$−DSPE content was reduced below 0.5 mol%; however, non-specific liposome binding increased. Thus, 0.5 mol% PEG$_{45}$−DSPE was used in favor of the initial 5 mol% in liposomal formulations moving forward.

Next, the relative amount of ganglioside (GM1, GM2, GM3, and GD1a) was titrated in the GLLs (Fig. 1c, Supplementary Fig. 3). A significant decrease in GLL binding to Siglec-1 CHO cells was observed when the ganglioside content exceeded a threshold of 1–3 mol% for GM1, GM2, and GD1a. In contrast, GM3 showed a different trend, with a progressive increase in GLL binding up to 20 mol% GM3. Ganglioside micelles (GM1) showed no binding to Siglec-1 (Supplementary Fig. 4). To rule out the possibility of different incorporation efficiencies into the liposomes for each ganglioside, we quantified ganglioside incorporation into the liposomes and found similarly high rates of incorporation for GM1[40], GM2, and GM3 (Supplementary Table 1). Because acyl chains of gangliosides can vary between species and tissues[20], we assessed the acyl chain structures and the binding of GM1 GLLs to Siglec-1 CHO cells using GM1 from four sources: bovine, ovine, porcine, and synthetic. No differences in binding between the different sources of GM1 to Siglec-1 were observed despite differences in their acyl chain structures (Supplementary Fig. 5). Cholesterol content was also varied and found to have a limited effect on GLL binding to Siglec-1 CHO cells (Supplementary Fig. 6). Conversely, the bulk lipid in the liposome formulation did impact GLL binding to Siglec-1 CHO cells, with 1-palmitoyl-2-stearoyl-sn-glycero-3-phosphocholine (PSPC) showing the highest binding (Supplementary Fig. 7). Accordingly, PSPC was used as the bulk lipid for the screening of the Siglec family.

### Exploring the ligand density dependency of Siglec binding

One explanation for the weaker binding of GM3 liposomes, compared to other GLLs, is that the intrinsic affinity of Siglec-1 for the GM3 trisaccharide is weaker than the oligosaccharide portion of the other gangliosides. To test this, we used a mass spectrometry-based Siglec binding assay[30] to determine the dissociation constant ($K_d$) of Siglec-1 to the oligosaccharide moieties of GM1, GM2, GM3, and GD1a (Fig. 1d, Supplementary Fig. 8). In contrast with the cell assay, the

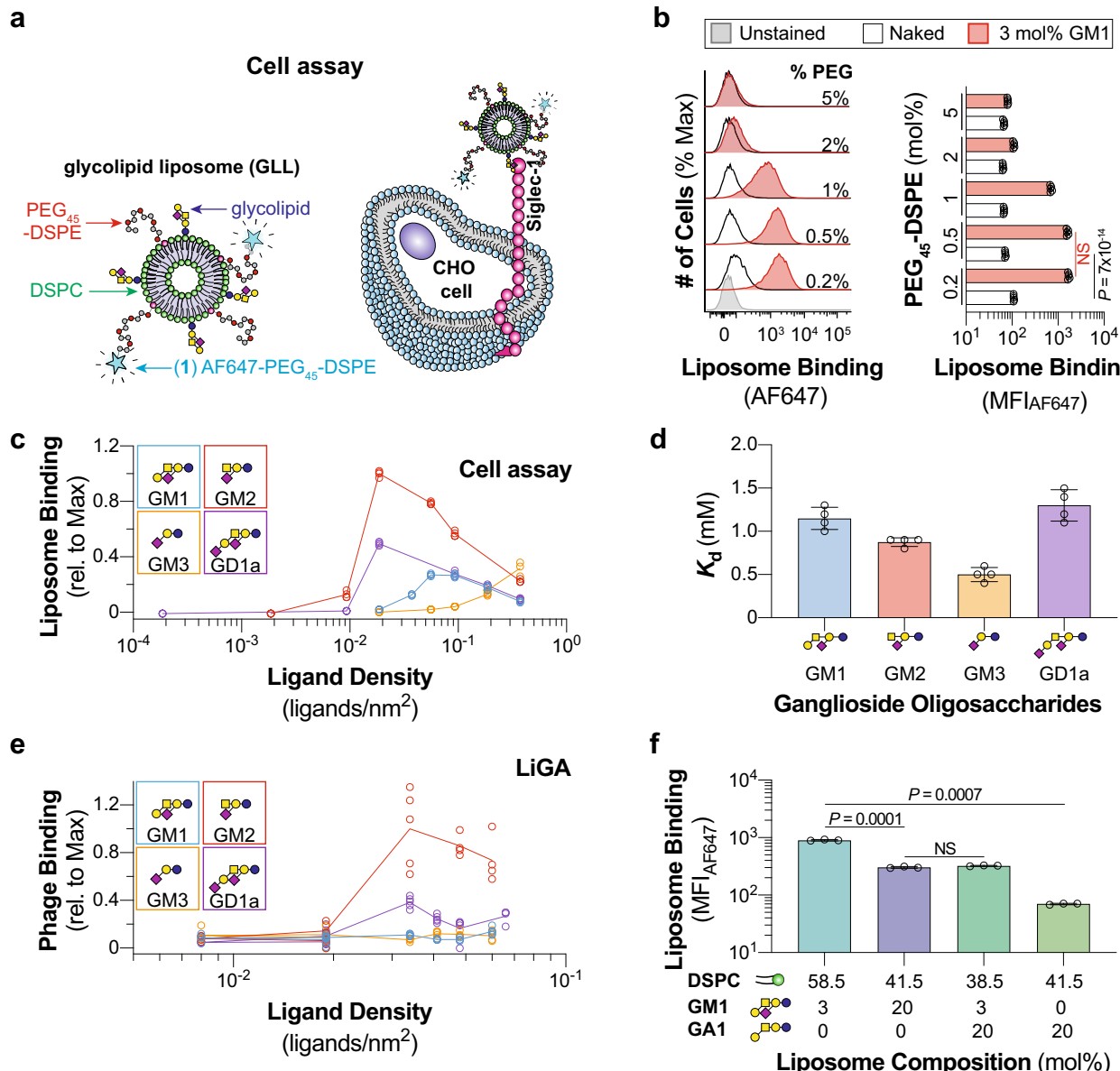

**Fig. 1 | Optimizing a liposome formulation for probing Siglec–ganglioside interactions. a** Schematic of the cell assay where binding between Siglec-1-expressing CHO cells and fluorescently labeled GLLs are quantified via flow cytometry. **b** Representative flow cytometry histograms and quantification of 3 mol% GM1 liposomes with varying mol% of PEG$_{45}$–DSPE to Siglec-1 CHO cells ($n = 5$ technical replicates). **c** Binding of liposomes formulated with increasing ganglioside (GM1, GM2, GM3, GD1a) content against Siglec-1 CHO cells ($n = 4$ technical replicates). **d** Mass spectrometry-derived dissociation constant for the interaction between soluble Siglec-1 and ganglioside (GM1, GM2, GM3, GD1a) oligosaccharides ($n = 4$ technical replicates). **e** Ganglioside (GM1, GM2, GM3, GD1a) oligosaccharide density titration using a liquid glycan array (LiGA) against Siglec-1 CHO cells ($5 \geq n \geq 4$ technical replicates). **f** Liposomes formulated with GM1 and GA1 binding to Siglec-1-expressing CHO cells in the cell assay ($n = 3$ technical replicates). Glycan structures are represented using Symbol Nomenclature for Glycans (SNFG; blue circle, glucose; yellow circle, galactose; yellow square, GalNAc; purple diamond, Neu5Ac). Data are represented as the mean ± one standard deviation of at least three technical replicates. For panels **b** and **f** a Brown–Forsythe and Welch one-way ANOVA was used for statistical analysis. Not Significant (NS), $P > 0.5$.

GM3 trisaccharide exhibited the highest affinity towards Siglec-1, with a $K_d$ value of $0.5 \pm 0.1$ mM. Therefore, factors beyond the intrinsic affinity of Siglec-1 to the oligosaccharide moiety of the ganglioside influence the avidity of Siglec-1 for glycolipids presented in a lipid bilayer.

We hypothesized that the unimodal density-dependent binding of Siglec-1 to GM1, GM2, and GD1a GLLs (Fig. 1c) is related to steric crowding[41]. To test this hypothesis, we used a liquid glycan array (LiGA)[42], wherein the oligosaccharide moieties of gangliosides were conjugated to a bacteriophage at different densities using a minimum of four independently barcoded preparations of the phage at each density, enabling binding to be read out by next-generation

sequencing of Siglec-1 CHO cells incubated with LiGA (Supplementary Fig. 9). For GM2 and GD1a, binding was maximal at a density of ~26 nm²/ligand (Fig. 1e). This optimal density was in a similar range as the optimal density of 18 nm²/ligand on liposomes. To further investigate molecular crowding, asialo GM1 (GA1) was added to the liposomal formulations. GA1 alone did not mediate binding to Siglec-1, yet excess GA1 (20 mol%) with GM1 (3 mol%) in the GLL formulation impaired binding to Siglec-1 compared to GM1 alone (Fig. 1f, Supplementary Fig. 10). In fact, liposomes with excess GA1 showed equivalent binding to that of high density (20 mol%) GM1 GLLs. As excess ganglioside densities are detrimental for Siglec-1 binding, 3 mol% gangliosides were used in GLL formulations moving forward. These results

suggest that the binding of a Siglec and a ganglioside in solution is not representative of the same interaction in a bilayer.

### Interrogating human Siglecs with ganglioside liposomes

Using the optimized liposomal formulation and our panel of CHO cells, the entire human Siglec family was tested against nine gangliosides in GLLs using untransfected (UT) CHO cells as a background. Binding of Siglec-1, -5, -6, -9, and -10 to multiple gangliosides was observed (Fig. 2a, Supplementary Fig. 11). Many novel interactions were observed, including Siglec-5 with GM1 and GD1b, Siglec-6 with numerous gangliosides, Siglec-9 with GM2, and Siglec-10 with GM1 and

GD3. Binding of GLLs to Siglec-1, -5, -9, and -10 was abrogated when their canonical essential arginine was mutated. Unexpectedly, mutation of the canonical essential arginine in Siglec-6 (Arg122) did not abrogate GLL binding. Although Siglec-6 has not been reported to bind gangliosides, Arg122 was reported as being critical for the recognition of sialylated ligands on cells[43]. These findings suggest that Siglec-6 has a secondary binding pocket capable of binding glycolipids.

It was surprising that Siglec-4 and -7 showed minimal GLL binding (Supplementary Fig. 11), given previous reports showing that GD1a and GT1b were ligands for Siglec-4, while GQ1b and GD3 were ligands for Siglec-7[44,45]. For Siglec-4, we considered that serum has ligands that

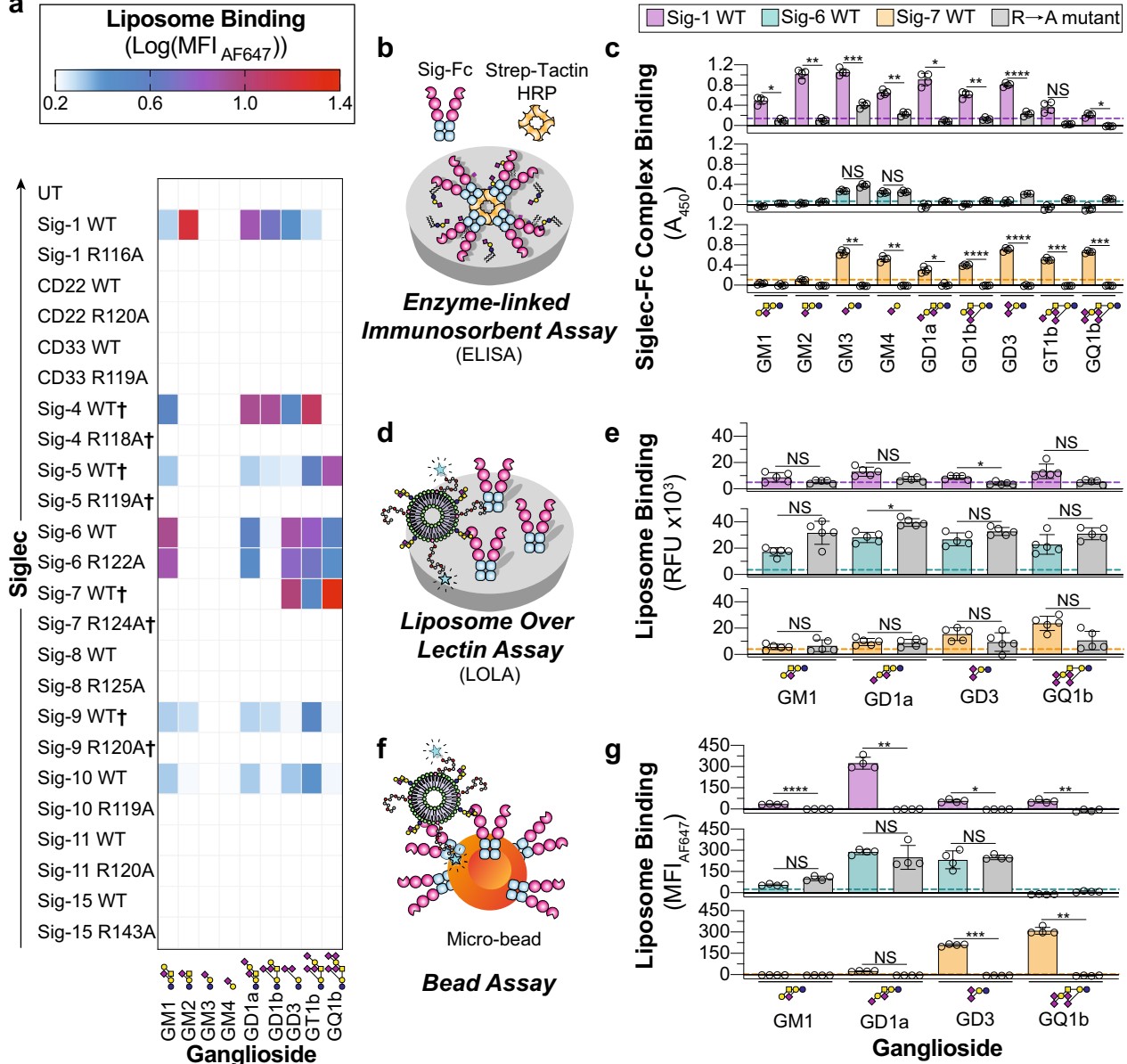

**Fig. 2 | Investigating Siglec–ganglioside interactions using optimized liposomes reveals interactions. a** Heatmap summarizing the binding interactions of each human Siglec ($4 \geq n \geq 3$ technical replicates). Color is representative of the $\log_{10}(\text{MFI}_{\text{AF647}})$ of each GLL subtracted from the $\log_{10}(\text{MFI}_{\text{AF647}})$ of the same GLL against UT CHO cells. † Denotes binding was measured after treatment of cells with neuraminidase. **b** Schematic of the enzyme-linked immunosorbent assay (ELISA) used to investigate Siglec–ganglioside interactions outside the context of a lipid bilayer. **c** ELISA results of Siglec-1, -6, and -7 binding to nine gangliosides ($n = 4$ technical replicates). **d** Schematic of the liposome over lectin assay (LOLA) where binding is read out on a fluorescence plate reader. **e** Results of the LOLA against a

select number of GLLs ($n = 5$ technical replicates). **f** Schematic of the bead assay where binding is read out by flow cytometry. **g** Results of the bead assay against a select number of GLLs ($n = 4$ technical replicates). All results are represented as the mean ± one standard deviation of at least three technical replicates. The dotted line on the ELISA, LOLA, and bead assay results represents two standard deviations above the blank well background for the ELISA and a naked liposome for the LOLA and bead assay. For panels **c**, **e**, and **g**, a Brown–Forsythe and Welch one-way ANOVA was used. Not Significant (NS), $P > 0.5$; *$0.05 > P \geq 0.01$; ** $0.01 > P \geq 0.001$; ***$0.001 > P \geq 0.0001$; ****$P < 0.0001$.

compete for binding with GLLs[36]; however, the absence of serum minimally impacted binding (Supplementary Fig. 12). We considered that these Siglecs may be masked by cis ligands on the CHO cells, preventing interactions with GLLs. Therefore, neuraminidase was used to remove cell surface sialic acid residues and 'unmask' the Siglec, which revealed the binding of GLLs to both Siglec-4 and -7. Unmasked conditions also modestly enhanced GLL binding to Siglec-5 and -9 (Fig. 2a, Supplementary Fig. 13). As ligand content was important for GLL binding to Siglec-1, we also titrated the amount of GM1 and GD3 in GLLs and tested binding against Siglec-6 and -7, respectively (Supplementary Fig. 14). Similar to Siglec-1, binding of GLLs to Siglec-6 and -7 decreased above an optimal ganglioside content.

## Oligosaccharide presentation influences Siglec-ganglioside binding

A previous study did not observe interactions between Siglec-6 and gangliosides in an Enzyme-Linked Immunosorbent Assay (ELISA)[46], suggesting that Siglec-6 may be tailored for binding glycolipids embedded in a membrane. To examine Siglec–glycolipid interactions further, we employed three additional assays. In these assays, we evaluated Siglec-1 and -7 together with Siglec-6, as Siglec-1 and -7 are both established ganglioside binders and have been shown to bind gangliosides outside of a bilayer[10,13]. The first assay was an ELISA, where gangliosides were adsorbed to a microplate and probed with our recently developed soluble dimeric Siglec-Fc proteins, bearing a Strep-tag II for precomplexing with tetrameric Strep-Tactin-Horseradish Peroxidase (HRP)[30] (Fig. 2b). Siglec-1 and -7 showed ganglioside binding in the ELISA that was similar to the cell assay (Fig. 2c). On the other hand, Siglec-6 did not recognize the gangliosides that it interacted with in the cell assay (GM1, GD1a, GD3, GT1b, and GQ1b), yet interacted weakly with GM3 and GM4. The same binding profile was observed with R122A Siglec-6. In the second assay, we developed a Liposome Over Lectin Assay (LOLA), wherein a Siglec-Fc is adsorbed to a microplate, followed by probing with the same fluorescently labeled GLLs used in the cell assay (Fig. 2d, Supplementary Fig. 15). All three Siglecs showed similar binding in the LOLA compared to the cell assay (Fig. 2e). Once again, binding was not abrogated in R122A Siglec-6. Finally, a bead assay was used wherein the Siglec-Fc was immobilized on streptavidin microbeads and probed for binding to GLLs by flow cytometry (Fig. 2f). Similar binding profiles were observed with the LOLA and cell assay (Fig. 2g, Supplementary Fig. 16). These biochemical assays provide further support that Siglec-6 uniquely requires presentation of gangliosides from a lipid bilayer for optimal engagement.

## Probing glycolipid recognition of Siglec-6 with synthetic neoglycolipids

To probe the specificity of Siglec-6 for glycolipids beyond the set of commercially available gangliosides, we prepared a panel of neoglycolipids (nGLs) and systematically varied the oligosaccharide, head-group/linker, and acyl chains (Fig. 3a). Focusing on the glycan moiety, nGLs 3 and 4 were prepared, which feature the oligosaccharide moiety of GM1 and GM3, respectively, linked to 1,3-di-O-hexadecyl glycerol via an amide linker (Fig. 3b). Neoglycolipid liposomes (nGLLs) formulated with 3 mol% 3 bound minimally to Siglec-6 CHO cells, whereas liposomes containing 3 mol% 4 showed a five-fold increase in binding compared to native GM3 GLLs (Fig. 3c). As the oligosaccharide of GM3 (α-Neu$p$5Ac-(2→3)-β-lactose) was preferred in this artificial presentation, the chemical diversity of our nGL panel was increased by linking the GM3 oligosaccharide to three different lipid groups by coupling the β-azidoethyl glycoside of α-Neu$p$5Ac-(2→3)-β-lactose to three different lipids through a triazole linkage to form nGLs 5, 6, and 7 (Fig. 3d). Liposomes formulated with 3 mol% 5 displayed relatively high binding to Siglec-6, whereas 6 and 7 showed lower binding (Fig. 3e, Supplementary Fig. 17). Using the triazole-linked di-O-hexadecyl

glycerol nGL scaffold, we further explored the glycan specificity of Siglec-6 with respect to regiospecificity by synthesizing three nGLs with triazole-linked α-(2→6)-sialyl-lactose (8), α-(2→3)-sialyl-LacNAc (9), and α-(2→6)-sialyl-LacNAc (10) (Fig. 3f). Compared to liposomes with 3 mol% 5, both WT and R122A Siglec-6 bound minimally to the conjugates containing an α-(2→6)-linked sialoside (Fig. 3g, Supplementary Fig. 18). Binding was also modestly lower when the lactose moiety of 5 was replaced with LacNAc (9). Binding of these nGLLs to R122A Siglec-6 strongly suggests that they target the same non-canonical glycolipid binding pocket on Siglec-6. As nGL 5 was used for targeting Siglec-6 in subsequent experiments, the content of 5 in liposomes was titrated and it was found that 5 mol% is optimal (Supplementary Fig. 19). To understand how 5 nGLLs engage Siglec-6, we performed a competition assay between GM1 GLLs and 5 nGLLs. The binding of GM1 GLLs decreased as the concentration of 5 mol% 5 nGLLs increased, suggesting that the two ligands compete for the same binding pocket (Supplementary Fig. 20). The strong ability of 5 to engage Siglec-6 was unique to a liposomal display, as 5 and GM3 showed only a 1.5-fold difference in binding to Siglec-6 by ELISA compared to the 34-fold difference in the cell assay (Supplementary Fig. 21).

## Probing the location of the noncanonical glycolipid binding pocket on Siglec-6

Recognition of glycolipids by Siglec-6 in a way that is independent of its canonical essential arginine prompted an investigation into how Siglec-6 interacts with glycolipids. We first tested if sialic acid is required by formulating GLLs with a variety of asialo glycolipids and found that none were able to mediate binding to Siglec-6 CHO cells (Supplementary Fig. 22). To explore how Siglec-6 recognizes sialosides independent of its canonical arginine residue, a molecular model of Siglec-6 was generated using AlphaFold[47] (Fig. 4a). The model's prediction was consistent with other Siglecs that have been crystalized, showing an extended conformation of their extracellular domains and a disulfide bridge between the V-set and underlying C2 domain[4].

We first investigated what domain(s) of Siglec-6 are required for glycolipid recognition by creating a series of chimeric Siglecs that consisted of different combinations of extracellular domains of Siglec-6 and Siglec-8. Siglec-8 was chosen because it has 46% sequence identity with Siglec-6 but does not bind GLLs. Surprisingly, none of the three individual Siglec-6 domains supported binding to liposomes formulated with 5 in the cell assay; however, when the first two domains of Siglec-6 were used, binding to 5 nGLLS was significantly above background (Fig. 4b, Supplementary Fig. 23a, b). A similar binding pattern was observed between the Siglec-6/8 chimeras and GD1a GLLs (Supplementary Fig. 23c). Modestly reduced binding of the chimeric construct consisting of the first two domains of Siglec-6, relative to WT Siglec-6, was likely due to a 76% reduction in expression of this construct (Supplementary Fig. 24). Indeed, when expressed as a soluble Fc conjugate and used in the LOLA, this construct displayed comparable levels of binding to WT Siglec-6 (Supplementary Fig. 25). These results suggest that both the V-set and the first C2 domain are needed for binding. In the AlphaFold model of Siglec-6, two cysteine residues, Cys46 and Cys172, form an interdomain disulfide bridge, which creates the interdomain interface (Fig. 4c). Despite even higher levels of expression than WT Siglec-6, the C46A and C172A mutants of Siglec-6 did not recognize 5 nGLLs. (Fig. 4d, Supplementary Fig. 26a, b). These results provide further evidence that the noncanonical glycolipid binding pocket of Siglec-6 requires an intact presentation of its first two extracellular domains.

The guanidinium functional group within an arginine residue commonly supports sialic acid binding, even for lectins outside of the Siglec family[48]. Accordingly, nine additional arginine residues in the V-set domain and at the interface of the V-set/C2 domain were mutated. Several mutants (R109A and R114A) displayed a modest reduction

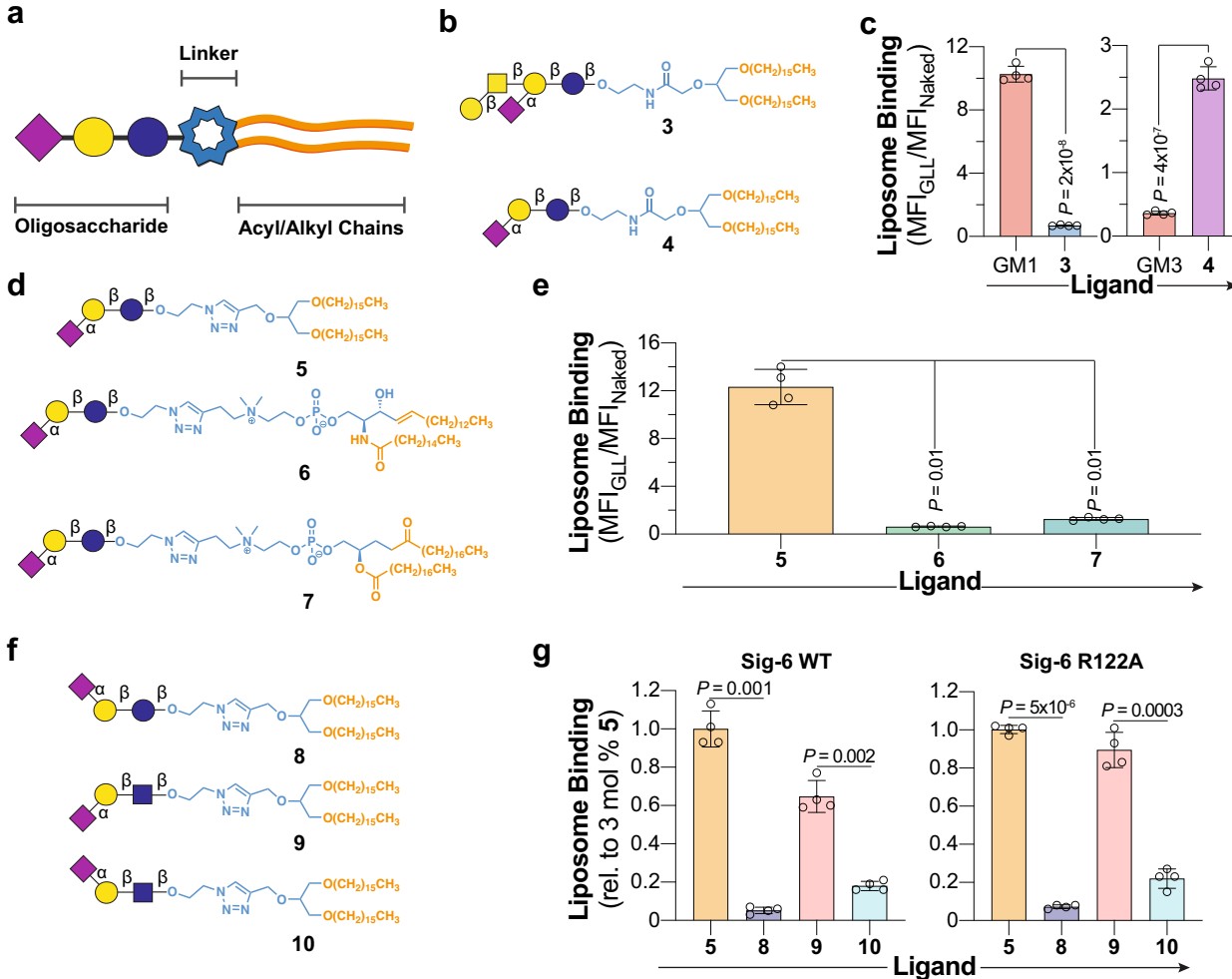

**Fig. 3 | Exploring the glycolipid binding specificity of Siglec-6 using nGLLs.**
**a** Schematic of a glycolipid structure broken down into three components.
**b** Structures of nGLs **3** and **4** presenting the oligosaccharide of GM1 and GM3, respectively, through an amide-linkage to a 1,3-di-*O*-hexadecyl glycerol scaffold. **c** Binding of liposomes formulated with 3 mol% **3** and **4** to WT Siglec-6 CHO cells in the cell assay, relative to liposomes formulated without a ligand ($n = 4$ technical replicates). **d**, Structures of nGLs **5**, **6**, and **7** presenting the oligosaccharide of GM3 triazole-linked to 1,3-di-*O*-hexadecyl glycerol, phosphatidyl sphingomyelin, and distearoylphosphatidylcholine scaffold, respectively. **e** Binding of liposomes formulated with nGLs **5**, **6**, and **7** to WT Siglec-6 CHO cells in the cell assay relative to

naked liposomes ($n = 4$ technical replicates). **f** Structures of nGLs **8**, **9**, and **10** presenting an α-(2 → 3)- or α-(2 → 6)-linked sialoside on an underlying lactose or LacNAc core, triazole-linked to 1,3-di-*O*-hexadecyl glycerol scaffold. **g**, Binding of liposomes formulated with **5** and **8**–**10** to WT and R122A Siglec−6 CHO cells in the cell assay relative to liposomes formulated with 3 mol% nGL **5** ($n = 4$ technical replicates). Data is representative of the mean ± one standard deviation of four technical replicates. For panel **c**, a two-tailed Student's *t*-test was used for statistical analysis. For panels **e** and **g**, a Brown–Forsythe and Welch one-way ANOVA was used for statistical analysis.

in binding to **5** nGLLs. A more pronounced reduction in binding was observed for the R92A and R100A mutants, albeit both were expressed at low levels compared to WT Siglec-6. (Fig. 4d, Supplementary Fig. 26a, b). As the R92A mutant showed minimal expression, a more conservative R92K mutant was used and gave cell surface expression levels at 42% compared to WT Siglec-6 (Supplementary Fig. 26c), but only minimal binding to **5** nGLLs (Fig. 4d, Supplementary Fig. 26d). To be confident that this residue was important for binding, careful gating was performed on equivalent levels of Siglec-6 expression between WT and R92K Siglec-6 and minimal binding was still observed with R92K Siglec-6 to **5** nGLLs (Supplementary Fig. 26e). An Fc chimera of R92K Siglec-6 was made and used in the bead assay, which demonstrated less than 5% of the binding to **5** nGLLs compared to WT Siglec-6 (Supplementary Fig. 27). Contributions from amino acids surrounding Arg92 were also investigated and F93A, L95A, and G175M mutants showed significantly decreased binding to **5** nGLLs (Fig. 4e, Supplementary Fig. 28). Notably, Gly175 is present within a loop in the underlying C2 domain that is predicted to protrude into the

interdomain cleft. These mutants further support that the interface between the V-set and the first C2-domain is important for glycolipid recognition.

## Targeting Siglec-6 on human cells and tissues

The strong binding of **5** nGLLs to Siglec-6 prompted an investigation into their use for targeting Siglec-6 on physiologically relevant cells. Siglec-6 has an unusual expression pattern and is found on mast cells, memory B-cells, placental syncytiotrophoblasts, and has no mouse ortholog[1]. As primary cells lack a genetic control for Siglec-6, we tested if pretreatment of cells with the anti-Siglec-6 antibody could be used to block **5** nGLL binding to Siglec-6 expressing CHO cells and found that it could (Supplementary Fig. 29). We first investigated if mast cells could be targeted by **5** liposomes through Siglec-6. Mast cells from human spleen samples show robust Siglec-6 expression levels but are in very low abundance (~0.01% of total white blood cells) (Supplementary Fig. 30). Siglec-6 levels on LAD2 cells, a mast cell line[49], were comparable to the primary mast cells (Fig. 5a). We found that **5** nGLLs

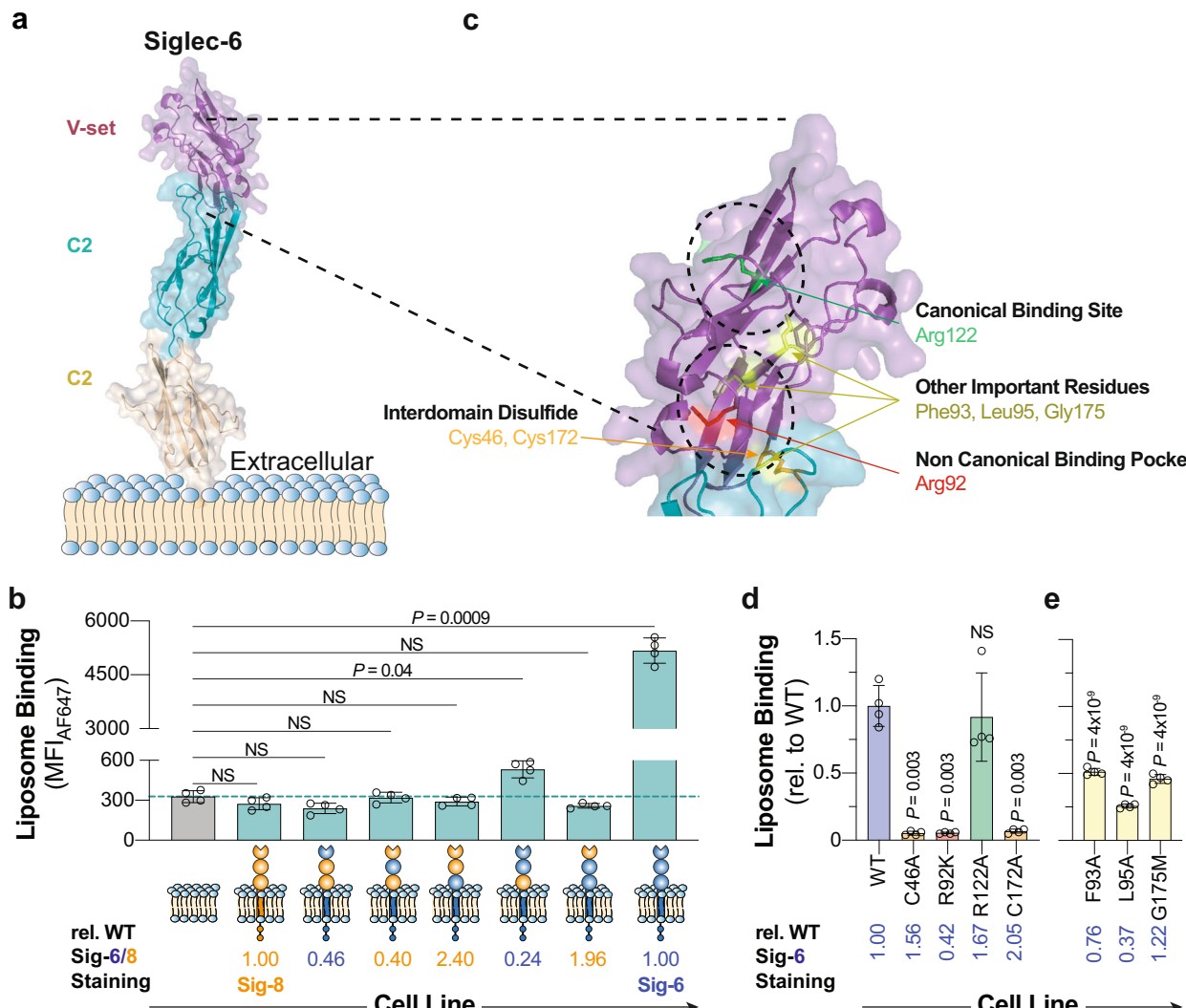

**Fig. 4 | Identifying the location of the noncanonical glycolipid binding pocket in Siglec-6. a** AlphaFold-generated molecular model of Siglec-6 extracellular domains. **b** Binding of **5** nGLLs to CHO cells expressing each Siglec-6/8 chimera. The dotted line represents the average binding of **5** nGLLs to UT CHO cells ($n = 4$ technical replicates). Relative Siglec staining is defined by the amount of anti-Siglec staining of the chimeric cell line over the anti-Siglec staining of the WT cell line. **c** Expansion of the molecular model of Siglec-6 at the interface of the V-set domain and the first C2 domain, with residues of interest highlighted as sticks (green,

canonical arginine residue; red, noncanonical arginine residue; orange, cysteine residues involved in the interdomain disulfide bridge, non-arginine residues; yellow). **d** and **e** nGLLs formulated with **5** binding to CHO cells expressing Siglec-6 mutants ($n = 4$ technical replicates). Liposome binding data is representative of the mean ± one standard deviation of four technical replicates. For panels **b** and **d**, a Brown–Forsythe and Welch one-way ANOVA was used for statistical analysis. For panels **d** and **e**, a statistical comparison was between each mutant and WT Siglec-6. Not Significant (NS) $P > 0.5$.

---

bound robustly to LAD2 cells, which was abrogated by pre-incubation with a Siglec-6 antibody (Fig. 5b, Supplementary Fig. 31).

We next tested the binding of **5** nGLLs against memory B-cells from peripheral human blood, using naïve B-cells as a control due to minimal Siglec-6 expression on this subset. From four healthy donors, we observed significant binding of the **5** nGLLs to memory B-cells compared to the naïve B-cells (Fig. 5c, Supplementary Fig. 32). Blocking of **5** nGLLs to human memory B-cells was achieved by pre-treatment with the anti-Siglec-6 antibody (Fig. 5d).

We also investigated whether **5** nGLLs could engage Siglec-6 expressed on syncytiotrophoblasts. Working with live human placental explant tissue cultures, syncytiotrophoblasts could be identified by the distinct pattern of phalloidin staining (Supplementary Fig. 33), we showed that **5** nGLLs are strongly associated with the syncytio-trophoblasts compared to liposomes without the nGL (Fig. 5e). In four independent biological replicates, **5** nGLLs showed significantly more puncta on these cells, suggesting that nGLLs bind to Siglec-6 on the

syncytiotrophoblasts (Fig. 5f). Consistent with this, **5** nGLLs showed a strong colocalization with Siglec-6 compared to naked liposomes (Fig. 5g). Moreover, pre-treatment with an anti-Siglec-6 antibody blocked these interactions (Fig. 5h). These results demonstrate that the glycolipid binding pocket on Siglec-6 can be targeted on primary human cells and tissues using synthetic epitopes.

### Extracellular vesicles (EVs) are bound and internalized by Siglec-6

EVs are reported to carry Siglec ligands including an abundance of sialylated glycolipids[50,51]. Therefore, we tested if EVs could interact with Siglec-6 by isolating and characterizing EVs from human peripheral blood (Supplementary Fig. 34a). EVs were fluorescently labeled and robust binding to Siglec-6 CHO cells was observed (Supplementary Fig. 34b), which was abrogated using the anti-Siglec-6 blocking antibody. (Fig. 6a, Supplementary Fig. 35). EVs also showed a similar binding profile to Siglec-6 as nGLLs, including equivalent binding to

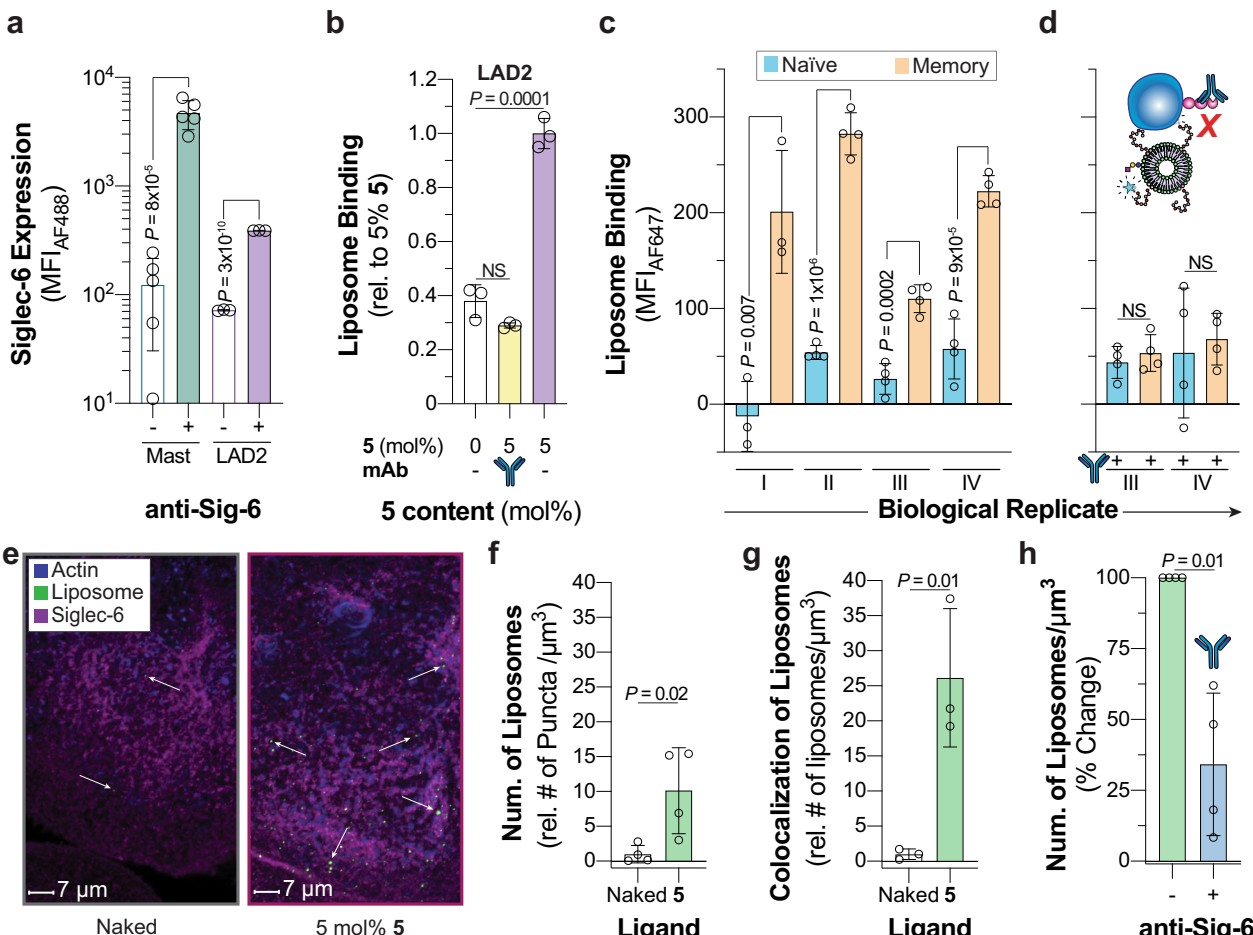

**Fig. 5 | Targeting Siglec-6 on human cells and tissues. a** Siglec-6 expression levels on primary mast cells isolated from healthy human spleen donors and LAD2 cells. For the primary mast cells, each datum represents a biological replicant (*n* = 5 biological replicates). **b** Binding of **5** nGLLs to LAD2 after treatment with an anti-Siglec-6 antibody quantified by flow cytometry (*n* = 3 technical replicates). **c** and **d 5** liposome binding to human naïve and memory B-cells isolated from the blood of four healthy biological replicants (represented with Roman numerals) without and with pretreatment of Siglec-6 blocking antibody, respectively (4 ≥ *n* ≥ 3 technical replicates). **e** Representative confocal microscopy images of human syncytiotrophoblasts after treatment with naked and **5** nGLLs. Quantification of the total number of naked liposomes and **5** nGLLs binding to human placental explants per μm³ (**f** *n* = 4 biological replicates) and the number of **5** nGLL colocalized with Siglec-6 per μm³ (**g** *n* = 3 biological replicates). **h** Binding of **5** nGLLs to human syncytiotrophoblasts after treatment with an anti-Siglec-antibody (*n* = 3 biological replicates). For panels **f**, **g**, and **h** each datum is representative of a different donor, which introduces variability between panels. All results are represented by the mean ± one standard deviation of at least three technical replicates. For panels **a**, **c**, **d**, **f**, **g**, and **h**, a two-tailed Student's *t*-test was used for statistical analysis. For panel **b**, a Brown–Forsythe and Welch one-way ANOVA was used for statistical analysis. Not Significant (NS) *P* > 0.5.

R122A Sigelc-6 and no binding to the C46A, C172A, and R92K mutants (Fig. 6b, Supplementary Fig. 36a, b). To ensure that different expression levels of Siglec-6 were not responsible for the observed decrease in binding in the cell assay, we used the bead assay and observed robust binding to WT Siglec-6 and no binding to R92K Siglec-6 (Fig. 6c, Supplementary Fig. 37). These results suggested that EVs bind to Siglec-6 in a similar manner as GLLs. In support of this, **5** nGLLs competed away binding of EVs to Siglec-6 (Fig. 6d, Supplementary Fig. 38). A modest, but significant, reduction in EV binding with EVs from two different donors was observed to LAD2 cells blocked with an anti-Siglec-6 antibody (Fig. 6e, Supplementary Fig. 39). Taken together, these results demonstrate that physiologically relevant expression levels of Siglec-6 support EV binding and that EV binding takes place through the interface between the V-set and the first C2 domain of Siglec-6.

After investigating how Siglec-6 engages EVs, we looked into how the properties and composition of the EVs influence binding to Siglec-6. First, we pre-treated EVs with a broadly acting neuraminidase A (Neu A) or an α-(2 → 3)-specific neuraminidase S (Neu S) to determine if Siglec-6 binds α-(2 → 3) linked sialosides as it did in the neoglycolipid

profiling (Fig. 6f, Supplementary Fig. 40). Indeed, significantly decreased EV binding was observed after treatment with either neuraminidase. These results suggest that EV binding is sialic acid-dependent and that α-(2 → 3)-linked sialosides on EVs mediate binding to Siglec-6. To examine whether gangliosides in EVs mediate binding to Siglec-6, we prepared EVs from WT and β1-4GalNT1⁻/⁻ N2a cells, as these knockout cells cannot synthesize complex gangliosides, which are the ligands for Siglec-6 (Supplementary Fig. 41). A significant reduction in binding of β1-4GalNT1⁻/⁻-derived EVs compared to WT EVs was observed (Fig. 6g), suggesting that complex glycolipids in EVs support binding to Siglec-6.

As Siglecs are endocytic receptors[52], we investigated if Siglec-6 can internalize cargo such as EVs. Using Daudi cells, a human B-cell line, transduced to overexpress Siglec-6, we assessed internalization of **5** nGLLs formulated with a pHrodo-labeled lipid (**13**), which increases fluorescence once within the acidic endosomal compartment, thereby reporting on cellular internalization[53]. A time-dependent increase in pHrodo signal for **5** nGLLs was observed independent of Arg122 and only at 37 °C, which is indicative of cellular internalization (Supplementary Fig. 42a). Imaging flow cytometry more directly revealed

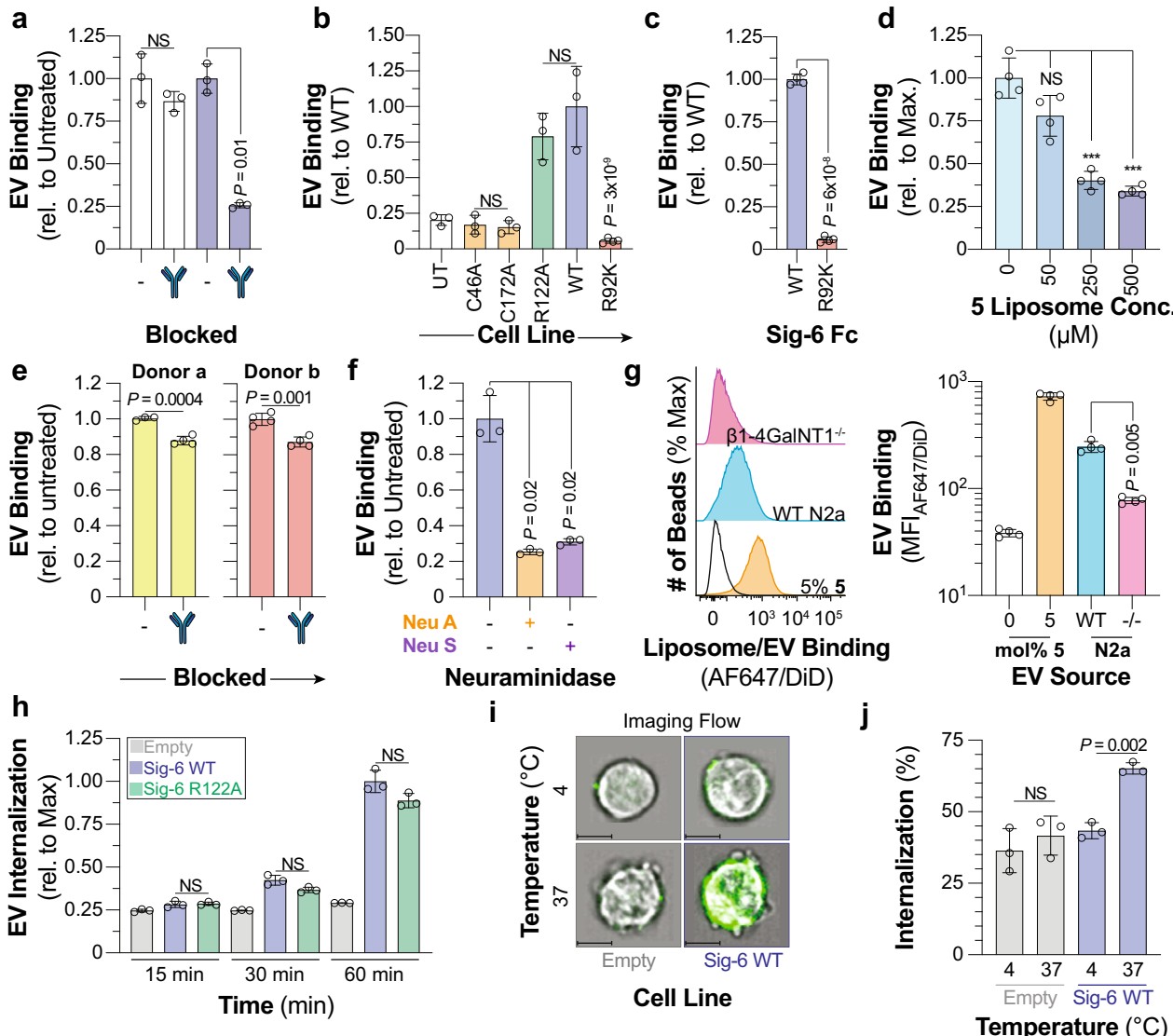

**Fig. 6 | Siglec-6 binds and internalizes extracellular vesicles through glycolipids independent of its conserved arginine residue. a** Binding of EVs to Siglec-6 expressing CHO cells pre-treated with an anti-Siglec-6 antibody ($n = 3$ technical replicates). **b** Binding of EVs to UT CHO cells and CHO cells expressing C46A, C172A, WT, R122A, and R92K Siglec-6 ($4 \geq n \geq 3$ technical replicates). **c**, Binding of EVs to WT and R92K Siglec-6 in the bead assay ($n = 4$ technical replicates). **d** Blocking of EV binding to Siglec-6 with 5 mol% **5** nGLLs in the bead assay ($n = 4$ technical replicates). **e** Binding of EVs isolated from two different donors to LAD2 cells pretreated with an anti-Siglec-6 antibody ($n = 4$ technical replicates). **f** Binding of neuraminidase A and neuraminidase S treated EVs to Siglec-6 in the bead assay ($n = 3$ technical replicates). **g** Binding of EVs isolated from WT and β1-4GalNT1$^{-/-}$ N2a cells to WT Siglec-6 in the bead assay ($n = 4$ technical replicates). **h** Time-

dependent fluorescence of pHrodo labeled EVs incubated with Daudi cells transduced with WT, R122A Siglec-6, and an empty vector ($n = 3$ technical replicates). **i** Representative imaging flow cytometry images of empty vector and WT Siglec-6 virally transduced Daudi cells incubated with AF488 labeled EVs at 4 or 37 °C with the EV fluorescence overlaid over the brightfield image. Scale bars represent 7 μm. **j** Quantification of internalization of EVs at 4 or 37 °C by Daudi cells transduced with WT Siglec-6 and an empty vector ($n = 4$ technical replicates). Data is represented by the mean ± one standard deviation of at least three technical replicants. For panel **a** (WT, C46A, R122A, C172A), **b**, **d**, **f**, **g**, **h**, and **j** a Brown–Forsythe and Welch one-way ANOVA was used for statistical analysis. For panels **a** (WT vs. R92K) **c** and **e** a two-tailed Student's $t$-test was used for statistical analysis. Not Significant (NS), $P > 0.5$.

that Siglec-6 Daudi cells internalize **5** nGLLs at 37 °C but not 4 °C (Supplementary Fig. 42b). Siglec-6 also mediated internalization of EVs, demonstrated by the time-dependent increase in pHrodo signal in WT and R122A Siglec-6 Daudi cells compared to Daudi cells transduced with an empty vector (Fig. 6h, Supplementary Fig. 43). Moreover, internalized EVs were observed by imaging flow cytometry at 37 °C (Fig. 6i, j). Therefore, Siglec-6 engages EVs through the glycolipid binding pocket and mediates their internalization.

## Discussion

Sialic acid-containing glycoproteins and glycolipids play many biological roles by serving as ligands for sialic acid-dependent lectins, such

as the Selectins and Siglecs. Gangliosides are highly abundant on mammalian cell membranes but studying the interactions between proteins and glycolipids in a biological membrane has many challenges. In addition to the complex mixture of glycolipids in a cell membrane, membrane dynamics, and composition can influence the conformation of the carbohydrate headgroup of gangliosides[40,54,55]. Here, we reduced this complexity by using liposomes displaying glycolipids to study their ability to engage immunomodulatory Siglecs. By developing an understanding of how the composition of the lipid bilayer influences the ability of Siglecs to engage gangliosides, we developed an optimal liposome formulation leading to the discovery of new Siglec–ganglioside interactions.

In the process of optimizing the liposomal formulation, a striking observation was the unimodal nature of the binding between Siglec-1 and GLLs formulated with GM1, GM2, and GD1a, which was not observed for GM3. Previous work examining GM3 content in liposomes and its impact on recognition by Siglec-1 found that binding increased up to 5 mol%, but higher ganglioside content was not reported[28]. This phenomenon was not unique to Siglec-1, as Siglec-6 and -7 also showed similar binding patterns with GM1 and GD3, respectively. Several lines of evidence in our studies, including the use of excess GA1 in the GLL formulation and the similar unimodal binding observed in the LiGA experiments, suggest that this behavior is due to steric crowding at high ganglioside content that negatively impacts binding. Steric crowding may arise from ganglioside–ganglioside interactions, which are dependent on the structure of the oligosaccharide[56], as a form of phase separation[57]. The divergent behavior of GM3 may be related to observations made in molecular modeling studies, which showed that the relatively small, linear GM3 trisaccharide is buried in the bilayer, making it difficult for Siglec-1 to access[58]. In line with this, we found that Siglecs-1, -6, and 7 do not efficiently engage GM3 when presented from a bilayer yet do engage GM3 outside of a bilayer as seen in the ELISA. The presentation of the oligosaccharide with respect to the bilayer may explain why genuine GM3 GLLs did not engage Siglec-6 whereas **5** nGLLs did. We speculate that the combination of the triazole-linkage and the di-*O*-hexadecyl glycerol lipid anchor more optimally presents the trisaccharide from the bilayer for engagement by Siglecs compared to genuine GM3 which is likely buried in the bilayer. However, the scope of our panel was not wide enough to resolve the contribution of each component to the presentation of the oligosaccharide. Except for GM3 and GM4, gangliosides presented from a lipid bilayer demonstrated remarkable avidity for Siglecs. For example, the soluble equivalent of **2** was previously found to have an $IC_{50}$ value of 0.38 μM with Siglec-1[38], which is over 1000-fold stronger than the affinity we report ($0.9 \pm 0.1$ mM) for GM2 to Siglec-1. Yet, when presented in a liposome at a similar ligand content, both **2** and GM2 have a similar ability to engage Siglec-1. This discrepancy may be related to **2** being presented at the end of a long PEG linker, leading to entropic costs in organizing multiple copies of the ligand for multivalent engagement of Siglec-1[59].

The effect of glycan density on Siglec binding has been well established[60] and in line with this, all the approaches used in this work, with the exception of the mass spectrometry-based Siglec binding assay, leverage avidity. Using the ELISA, LOLA, cell, and bead assays many of the established Siglec-ganglioside interactions were reproduced, specifically with Siglec-1, -4, -5, -7, -9, and -10[10,12,18]. In addition, we found interactions including: Siglec-4 with GM1, GD1b, and GD3; Siglec-5 with GM1; Siglec-7 recognizes GM4; Siglec-9 recognizes GM1; and Siglec-10 recognizes GM1, GM2, and GD3. However, not all experimental platforms revealed the same interactions, and it is important to consider how membrane dynamics influence avidity. There are discrepancies in the literature and this study with respect to Siglec–ganglioside interactions that are likely due to the experimental format used. For example, binding of Siglec-6 to glycolipids had never been observed, although Siglec-6 has only been investigated with an ELISA to GD3[43], and the lack of observed binding agrees with our ELISA results. However, in approaches where the ganglioside resides in a bilayer, robust binding of many gangliosides including GD3 was observed. In summary, the interpretation of these results with respect to biological significance requires careful consideration of the experimental platform.

In our studies, we demonstrated that Siglec-6 is proficient at binding and internalizing nGLLs and EVs independent of its canonical essential arginine. We identified the interface between the V-set domain and the first C2 domain as being critical for glycolipid recognition. It was previously reported that the canonical binding pocket in Siglec-6 recognizes α-(2 → 6)-linked sialosides[43]; however, our results

from the nGLL-binding assays as well as the neuraminidase S treatment of EVs suggest that this noncanonical binding pocket in Siglec-6 prefers α-(2 → 3) sialosides. Interestingly, Arg92 in Siglec-6 is completely conserved in other Siglecs[13], yet Siglec-6 was the only Siglec that could interact with glycolipids in this noncanonical manner. Previously, it was noted that Arg94 in Siglec-7, which aligns with Arg92 of Siglec-6, is involved in the recognition of disialylated glycolipids[44]. Although we do not rule out that Arg94 within Siglec-7 makes up a secondary binding pocket that is responsible for enhancing its binding to disialylated ligands, it should be noted that mutation of its canonical essential arginine (Arg124) completely abolished recognition of gangliosides in all our assays.

The ability of our optimal formulation combined with our ligand **5** to target this noncanonical binding pocket of Siglec-6 on physiologically relevant cells and tissues presents many therapeutic opportunities as Siglec-6 expression is relevant to several pathological conditions including acute myeloid leukemia[61], colorectal cancer[62], and preeclampsia[63]. While eliminating the majority of PEG from liposomes will affect their pharmacokinetics, gangliosides have been known to substitute for PEG in giving liposomes 'stealth' properties[64]. With the growing concern over PEGylated therapeutics, due to side effects caused by their ability to elicit anti-PEG antibodies[65], gangliosides displayed in liposomes are an excellent alternative and their ability to interact with Siglecs will be important to consider.

Our observation that ganglioside recognition on liposomes is shielded by PEG could be analogous to the glycocalyx of cells. Indeed, there are few examples of trans interactions between Siglecs on one cell and gangliosides on another[66]. These results point to other relevant biological locations where gangliosides may serve as Siglec ligands, such as on EVs and viruses. Gangliosides are found ubiquitously across all cell types and tissues and contribute to the composition of EVs[67]. Siglec-1[23] and -7[26,50] have both been shown to engage EVs in a manner that is predicted to be through glycolipids. The ability of Siglec-6 to recognize EVs is particularly relevant in the context of syncytiotrophoblasts of the placenta, which are bathed in maternal blood containing an abundance of maternal EVs[68]. Fetal–maternal immunological tolerance relies on communication between the fetus and mother[69], which Siglec-6 has the potential to participate in through recognition of maternal EVs. Similarly, the ability of Siglec-6 to be engaged by ligands on EVs may have important implications for helping to maintain immunological tolerance.

In conclusion, screening the entire human Siglec family against a panel of gangliosides presented in a lipid bilayer within an optimized liposome revealed many previously undiscovered Siglec–glycolipid interactions, most notably between Siglec-6 and several gangliosides. Noncanonical recognition of gangliosides by Siglec-6 is tailored for recognizing glycolipids in lipid bilayers. Probing this glycolipid binding pocket with a panel of synthetic nGLs yielded neoglycolipid **5**, which has greater avidity for Siglec-6 compared to natural gangliosides, enabling targeting of liposomes to Siglec-6-expressing cells and tissues, which opens future drug delivery applications. This glycolipid-binding pocket on Siglec-6 engages and internalizes EVs, demonstrating its utility as a versatile immunomodulatory receptor with the potential of participating in immunological tolerance at numerous cellular locations.

## Methods

### Human samples
All experiments involving human blood samples and placental sample collection were approved by the human research ethics board (HREB) biomedical panel at the University of Alberta.

### Cell culture and growth medium
CHO Flp-In cells (ATTC) were cultured in DMEM/F12 Media (Gibco) supplemented with 5% (V/V) fetal bovine serum (Gibco), penicillin

(100 U/mL), and streptomycin (100 μg/mL). Cells were grown at 37 °C and 5% $CO_2$ in tissue culture flasks (VWR).

## Cloning of Siglec constructs

The genes for Siglec-1-11 and 15 were synthesized by *GeneArt* (Thermo Fisher) and designed with a 5' *NheI* and 3' *AgeI* site immediately before the start codon and stop codon, respectively. When appropriate, silent mutations were introduced to remove internal *NheI* and *AgeI* cut sites. Each gene was cut out from the initial vector via a double restriction enzyme digest with *NheI* and *AgeI* (Thermo Fisher). Digestion was confirmed by running a 1% agarose gel after which the digested gene was excised from the gel and a gel purification was performed using a GeneJET Gel Extraction Kit (Thermo Fisher) to isolate the double-digested gene. The digested gene was ligated into *NheI* and *AgeI* digested pCDNA5 vector and then transformed into chemically competent *Escherichia coli* DH5α. Colonies were then picked and transferred to liquid culture (lysogeny broth plus 50 μg/mL ampicillin) and left to grow overnight in a shaking incubator at 37 °C. The plasmid was then isolated from the bacterial culture using GeneJET Plasmid Miniprep Kit (Thermo Fisher) and the successful incorporation of the Siglec gene into pCDNA5 was validated by restriction enzyme digestion and Sanger sequencing.

## Site-directed mutagenesis

Mutations were introduced into each Siglec via gene overlap extension PCR mutagenesis (primers were ordered through IDT). Mutagenesis primers can be found in Supplementary Tables 2 and 3 for the critical arginine mutants of the entire Siglec family and the additional Siglec-6 mutants respectively.

## Stable transfection

CHO Flp-In cells were initially cultured as described above. The transfection began by seeding 400,000 cells in a six-well cell culture dish. The next day, 0.2 μg of the desired Siglec DNA in pCDNA, 2 μg of pOG44 plasmid, and 7 μg of Lipofectamine Plus reagent (Thermo Fisher) were added to 250 μL of Opti-MEM (Gibco) and the mixture was incubated at room temperature for 15 min. Next, 8 μL of Lipofectamine LTX reagent (Thermo Fisher) was added to the mixture; and the mixture was left at room temperature for 30 min. During the incubation, the seeded cells were gently washed with Opti-MEM. The transfection mixture was then added to the seeded cells and the cells were left in the growing conditions described above overnight. The next day the media was aspirated from the well and replaced with CHO growth media (described above). The cells were then selected over 2 weeks by gradually increasing the amount of Hygromycin B from 0.5 to 1 mg/mL, replacing the media every other day.

## Liposome production

Stock lipids (DSPC, Cholesterol, PEG$_{45}$–DSPE) solutions were prepared by solvating the lipids in chloroform. An appropriate volume of each lipid solution was transferred to a glass test tube to reach the desired mol% of the component. The chloroform was then removed under $N_2$ gas to yield a lipid thin film. Once dry, 100 μL of DMSO was added to the lipid-containing test tube and then the functional lipids (ganglioside, AF647–PEG$_{45}$–DSPE, Siglec ligand–PEG$_{45}$–DSPE, pHrodo–PEG$_{45}$–DSPE, NGL) was added from their respect DMSO stock solution. The lipid mixture was then stored at −80 °C until completely frozen at which point excess DMSO was removed via lyophilization. Dry lipids were then stored at −80 °C until they were extruded. Lipid suppliers can be found in Supplementary Table 4.

## Liposome extrusion

Dry lipids were then hydrated in 1 mL of PBS. The lipids were then sonicated in a bath sonicator in a cycle of 1 min on, 5 min off for a total of five cycles. Liposomes were then extruded using an Avanti mini extruder first using an 800 nm filter and then a 100 nm filter yielding liposomes that were 130 ± 35 nm (Supplementary Table 5) measured using a Malvern Panalytical Zetasizer Nano S. Liposomes were stored at 4 °C.

## Cell assay

200,000 cells/well were plated into a 96-well U-bottom microplate and centrifuged at 300 × g for 5 min. The cells were then resuspended in 50 μL of 50 μM liposome/50 μg/mL (protein concentration) of EV solution in DMEM F12 5% FBS and incubated at 37 °C for 30 min. Excess liposomes were then removed via centrifuged at 300 × g for 5 min followed by washing the cells with FACS solution. The cells were then resuspended in (1:250 V/V) anti-Siglec-flow buffer (1% V/V FBS, 500 μM, EDTA Hank's Balanced Salt Solution pH 7.4), solution and left to rest on ice for 30 min. The cells were then centrifuged again at 300 × g for 5 min and the cells were resuspended in FACS buffer solution and were then ready for analysis by flow cytometry. Information regarding antibody clone, catalog number supplier, label isotype, and dilution can be found in Supplementary Table 6. The Siglec-positive cells were gated on for liposome-binding analysis (Supplementary Fig. 1b).

## Flow cytometry

Flow cytometry measurements were collected on a 5-laser Fortessa X-20 (BD Bioscience). All the resulting data were analyzed using FlowJo (10.5.3) software (BD Biosciences).

## Ganglioside content quantification

The average ganglioside content in a given liposome sample was measured by electrospray ionization mass spectrometry (ESI-MS) using an internal standard (IS)[70]. Briefly, the liposome sample was first disassembled in a methanol solution of 0.15% formic acid, and a known amount of IS was added. The solution was then analyzed by ESI-MS. The total ion abundance (Ab) ratio of the ganglioside (GSL) to the IS measured by ESI-MS is related to the corresponding solution concentration ratio by Eq. (1):

$$R = \frac{Ab(GSL)}{Ab(IS)} = \frac{[GSL]}{[IS]} \tag{1}$$

and thus the ganglioside concentration can be obtained from linear regression. The corrected ganglioside percentage (GSL%$_{corr}$) and the ganglioside incorporation in the liposome can be found from Eqs. (2) and (3), respectively, where [GSL]$_{nominal}$ is the nominal concentration of the ganglioside used for liposome preparation:

$$GSL\%corr = \frac{[GSL]}{[\text{Total Lipid}]} \tag{2}$$

$$\text{Incorporation} = \frac{[GSL]}{[GSL]_{nominal}} \tag{3}$$

The corresponding deuterium labeled gangliosides, N-ω-CD3-octadecanoyl monosialoganglioside GM2 (GM2-d3), and N-ω-CD3-octadecanoyl monosialoganglioside GM3 (GM3-d3), purchased form Matreya LLC (State College, PA), were used as IS for GM2 and GM3 content quantification, respectively. All these measurements were carried out in negative ion mode using a Q Exactive Hybrid Quadrupole-Orbitrap mass spectrometer coupled with the Nanospray Flex Ion Source (Thermo Fisher Scientific, Bremen, Germany). Tips pulled from a borosilicate capillary (1.0 mm o.d., 0.78 mm i.d.) by a micropipette puller (P-1000, Sutter Instruments, Novato, CA) were used to perform the nanoflow ESI. The sample solution was loaded into the nanoflow ESI tip and a voltage of −0.7 kV was applied to a platinum wire inserted into the tip and in contact with the sample solution. For the Orbitrap mass spectrometer, the key parameters were: capillary

temperature 160 °C, maximum inject time 100 ms, microscans 2, and resolution 140,000. All other parameters were set at default values. Data acquisition and processing were performed using Xcalibur (Thermo Fisher Scientific, version 4.4).

### Direct ESI-MS binding assay

The affinities of ganglioside (GM1, GM2, GM3, and GD1a) oligosaccharides (L) for the Siglec-1 fragment were measured by the direct ESI-MS assay[1]. A reference protein ($P_{ref}$) was added to the ESI solutions in order to correct mass spectra for any nonspecific binding that occurred during the ESI process[2]. The dissociation constant ($K_d$) was calculated from the total abundance (Ab) ratio ($R$) of the ligand-bound (PL) to free Siglec-1 fragment (P) ions (Eq. (4)) measured by ESI-MS for solutions of known initial concentration of Siglec-1 fragment ($P_O$) and ligand ($L_O$), Eq. (5):

$$R = \frac{Ab(PL)}{Ab(P)} = \frac{[PL]}{[P]} \tag{4}$$

$$K_d = \frac{[P][L]}{[PL]} = \frac{[L]_0}{R} - \frac{[P]_0}{R+1} \tag{5}$$

The reported $K_d$ values correspond to average values measured at 3.6 μM of Siglec-1 fragment and 20, 40, 80, and 140 μM of each ganglioside oligosaccharide tested. Direct ESI-MS measurements were performed with nanoflow ESI in positive ion mode (voltage -1 kV) on a Q-Exactive Orbitrap mass spectrometer (Thermo Fisher Scientific). Automatic gain control (AGC) target, the maximum inject time, capillary temperature, and S-lens RF level were set to $1 \times 10^6$, 100 ms, 150, and 100 °C, respectively. The resolution was 17,500 at $m/z$ 200. Data acquisition and processing were carried out using Xcalibur (Thermo Fisher, version 4.1).

### Modification of phage clones with glycans

Solution of five SDB phage clones ($10^{12}$–$10^{13}$ PFU/mL in PBS) was combined in equal amounts (50 μL) to create a multiplexed silent barcode (MSDB). Twenty-four such MSDB were created and combined DBCO-NHS (50 mM in DMF) to afford a final concentration ranging from 0.25 to 2 mM. The reaction was incubated for 45 min at room temperature. The conjugates were purified by Zeba™ Spin Desalting column (7 kDa MWCO, 0.5 mL, Thermo Fisher) and pVIII modification rate was confirmed by MAL/DI using a previously reported protocol. Typically, 1 mM DBCO-NHS yields 25% of pVIII modification after 45 min incubation. Six MSDB were combined and a solution of azido-glycans (10 mM stock in Nuclease Free $H_2O$) was added to the solution to afford a 2 mM concentration of glycan-azide and the solutions were further incubated overnight at 4 °C. The glycan conjugation was confirmed using MALDI-TOF. The conjugates were purified by Zeba column and supplemented with glycerol and stored as 50% glycerol stock at −20 °C. LiGA mixture was prepared by combining these solutions.

### Modification of phage clones with glycans

The solution of five SDB phage clones ($10^{12}$–$10^{13}$ PFU/mL in PBS) was combined in equal amounts (50 μL) to create a multiplexed silent barcode (MSDB). Twenty-four such MSDB were created and combined DBCO-NHS (50 mM in DMF) to afford a final concentration ranging from 0.25 to 2 mM. The reaction was incubated for 45 min at room temperature. The conjugates were purified by Zeba™ Spin Desalting column (7 kDa MWCO, 0.5 mL, Thermo Fisher) and the pVIII modification rate was confirmed by MAL/DI using a previously reported protocol. Typically, 1 mM DBCO-NHS yields 25% of pVIII modification after 45 min incubation. Six MSDB were combined and a solution of azido-glycans (10 mM stock in Nuclease Free $H_2O$) was added to the solution to afford a 2 mM concentration of glycan-azide and the solutions were further incubated overnight at 4 °C. The glycan

conjugation was confirmed using MALDI-TOF. The conjugates were purified by Zeba column and supplemented with glycerol and stored as 50% glycerol stock at −20 °C. LiGA mixture was prepared by combining these solutions.

### Binding of LiGA to CHO Cells expressing Siglec-1

Confluent CHO cells expressing Siglec-1 or UT CHO cells were detached using PBS plus 5 mM EDTA and washed with PBS ($2 \times 5$ mL). Suspension of at $2 \times 10^6$ cells in HEPES-1, % BSA (20 mM HEPES, 150 mM NaCl, 2 mM $CaCl_2$, pH 7.4, 1% BSA) in a round bottom 3 mL tube (Corning, #352054) was combined with LiGA ($10^8$ PFU) and incubated for 1 h at 4 °C. The cells were then washed with HEPES-0.1% BSA in ($2 \times 3$ mL) and HEPES buffer ($1 \times 1$ mL). The washed cell pellet was resuspended in 30 μL nuclease-free $H_2O$. The solution was incubated at 90 °C for 15 min, centrifuged at $21,000 \times g$ for 10 min, and 25 μL of the supernatant was used for PCR and sequencing.

### Liposome ligand density calculation

The total number of lipids in a liposome ($N_L$) was calculated using Eq. (6). Liposomes were assumed to be a sphere 100 nm in diameter ($d$). The thickness of a DSPC bilayer ($h$) was assumed to be 5 nm. The area of a phosphatidylcholine head group ($b$) was assumed to be 0.71 nm².

$$N_L = \frac{4\pi\left(\frac{d}{2}\right)^2 + 4\pi\left[\frac{d}{2} - h\right]^2}{b} \tag{6}$$

The ligand density (LD) was then calculated using Eq. (4) where $\chi$ is the appropriate mol% of ganglioside in the liposome formulation assumed that 50% is on the outer leaflet of the bilayer, $N_L$ was calculated using Eq. (7), $N_A$ is Avogadro's number and $d$ is the diameter of the liposome.

$$LD = \frac{\left(\frac{\chi}{2}\right)(N_L)(N_A)}{4\pi\left(\frac{d}{2}\right)^2} \tag{7}$$

### Unmasking cell assay

Cells were harvested and resuspended in complete media with either Neuraminidase A or S. The cells were then placed in a 37 °C shaking incubator for 1 h. The cells were then washed with complete media and then the liposome binding assay was performed as described above.

### Siglec-Fc production

Siglec-Fc constructs designed by Rodrigues et al.[30] were used in this work. Siglec-Fc constructs were expressed using WT CHO cells in cell culture media as described above with 1% HEPES. Cells incubated as described above for one-week post confluency. The supernatant was harvested and stored at 4 °C.

### Siglec-Fc purification

The purification was heavily inspired by Rodrigues et al.[30]. The purification of the Siglec-Fcs from supernatant began with $Ni^{2+}$ affinity chromatography using an AKTA FPLC equipped with a HisTrap Excel column (GE). The column was equilibrated with fifteen column volumes (CVs) of equilibrium buffer (500 mM NaCl, 20 mM $NaPO_4H_2$, pH 7.4). The Siglec-Fc containing supernatant was then loaded in its entirety onto the column. The column was then washed with fifteen CVs of wash buffer (500 mM NaCl, 30 mM imidazole, 20 mM $NaPO_4H_2$, pH 7.4). The Siglec-Fc was then eluted in 20 in 1 mL fractions with twenty CVs of elution buffer (500 mM NaCl, 500 mM imidazole, 20 mM $NaPO_4H_2$, pH 7.4). Fractions containing Siglec-Fc were compiled and then prepared for the next stage of purification by diluting the fractions 1:4 with buffer W (100 mM Tris–HCl, 150 mM NaCl, 1 mM EDTA, pH 8.0). A Strep-Tactin column (IBA life sciences) was washed with fifteen CVs of buffer W. The Siglec-Fc-buffer W solution was then

loaded in its entirety onto the column. The bound Siglec-Fc was then washed with fifteen CVs of buffer W. Siglec-Fc was eluted from the column with 15 column volumes of buffer E (100 mM Tris–HCl, 150 mM NaCl, 1 mM EDTA, 10 mM *d*-Desthiobiotin, pH 8.0). Fractions containing Siglec-Fc were pooled and dialyzed against PBS to remove any *d*-Desthiobiotin. After dialysis, Siglec-Fcs were concentrated using an ultra-centrifugal device (30 kDa MWCO) and aliquoted into 5 μg aliquots, and frozen. The Siglec-Fcs were then lyophilized overnight, and the lyophilized Siglec-Fcs were stored at −20 °C.

## Ganglioside ELISA
Our ELISA approach was carried out similarly to previous work by Rapoport[12] and Yamakawa[44]. Ganglioside ethanol solutions (10 μM) were prepared and then transferred in 50 μL increments to a 96-well microplate. The ethanol was removed by drying the plates overnight at room temperature. The plates were then washed with PBS, dried, and then blocked with 5% (m/V) BSA PBS for 1 h. The plates were then washed with PBS and then 2 μg/mL Siglec-Fc precomplexed to Strep-Tactin (2 Siglec-Fc:1 Strep-Tactin monomer) horse radish peroxidase was added to the microplate. The complex was incubated with the plate at room temperature for 2 h. The unbound complexes were then removed by washing in PBS and the plate was developed with Sera care TMB solution. The amount of binding was then quantified by using the background (microplate well with no ligand) subtracted absorbance at 450 nm using a Molecular Devices SpectraMAX ® iD5.

## Liposome over lectin assay (LOLA)
Siglec-Fc in PBS was adsorbed to a 96-well flat-bottom fluorescent microplate by adding 1 μg/well and incubating the plate at 4 °C overnight. The plate was then washed with PBS followed by blocking with 5% (m/V) BSA in PBS for 1 h at room temperature. The plate was again washed with PBS and 100 μM liposome in PBS was added to each well. The plate was then incubated at 37 °C for 30 min followed by washing with PBS. PBS was then added to the plate and the fluorescence intensity ($\lambda_{ex}$ 640 nm, $\lambda_{em}$ 680 nm) of each well was measured using a Molecular Devices SpectraMAX ® iD5.

## Bead assay
Pierce™ Streptavidin Magnetic Beads (Thermo Scientific) were blocked with 2% (m/V) BSA on ice for 1 h. Siglec-Fc in PBS was added then added to the bead solution such that the final concentration was 25 μg/mL. The Siglec-Fc was complexed to the beads for 1 h on ice. Excess Siglec-Fc was removed by washing the beads with 2% BSA solution and 50 μM liposome or 50 μg/mL (protein concentration) EV solution was then added to the beads. The beads were incubated with the liposomes/EVs for 30 min at 37 °C. The beads were then washed with 2% BSA solution and flow cytometry was used to measure the binding between the beads and the liposomes.

## Generation of Siglec-6/8 chimeric constructs
The genes for the constructs containing each domain of Siglec-6 to the two others from Siglec-8 were synthesized by *GeneArt* and designed with a 5′ *NheI* and 3′ *AgeI* site immediately before the start codon and stop codon respectively. When appropriate, silent mutations were introduced to remove internal *NheI* and *AgeI* cut sites. The additional two constructs containing the two domains of Siglec-6 next to each other were generated by gene overlap extension mutagenesis using the primers in Supplementary Table 7 and using the constructs above as a template.

## Isolation of white blood cells from human spleen
Spleen tissue was cut into pieces ~1 cm³ and placed into a petri dish containing RPMI supplemented with FBS (10% V/V) and penicillin (100 U/mL) and streptomycin (100 μg/mL) that was chilled to 4 °C. White blood cells were then separated from the rest of tissue using a

*Miltenyi gentleMACS Dissociator*. The tissue homogenate was then passed through a tea strainer and the filtrate was centrifuged at $400 \times g$ for 10 min. The supernatant was removed, and the cell pellet was resuspended in 4 °C red blood cell lysis solution (StemCell Technologies) and the cells were incubated for 10 min. The white blood cells in the lysis solution were then passed through a 75 μm cell strainer and the filtrate was then diluted 5-fold with supplemented RPMI and centrifuged for at $400 \times g$ for 10 min. The pellet was washed three more times with supplemented RPMI. Resuspend the cells in DMSO/FBS (1:9 V/V) and freeze cells using a Mr. Frosty™ at −80 °C for 2 days. The cells were then moved into liquid nitrogen and stored until needed.

## Liposome uptake assay in the placenta explants
Liposome uptake assay in the placenta explants was performed as described previously by Shaha et al.[71]. 6.5–7.5 weeks gestation placentas were obtained from elective pregnancy terminations after informed patient consent in accordance with methods approved by the University of Alberta Human Ethics Research Board. Whole placenta was rinsed in cold PBS and the placenta was cut into uniform 2 mm³ pieces and incubated overnight at 37 °C in Iscove's modified Dulbecco's medium (IMDM) supplemented with 10% (V/V) fetal calf serum. Following overnight incubation, placental explants were serum-starved in serum-free IMDM with 0.5% (m/V) bovine serum albumin and 25 mM HEPES buffering agent for 1 h then incubated in 50 μM liposome media solution (5% **5**, 0% liposome control) for 2 h and washed with cold PBS before fixation in 4% paraformaldehyde (PFA). For Siglec-6 blocking assays, following overnight incubation, placental explants were first blocked with human IgG (1:50 dilution; Thermo Fisher) for 30 min at 37 °C, and then, explants were incubated either with Siglec-6 antibody (1:50 dilution) or serum-free media for another 30 min at 37 °C. Following blocking, placental explants were incubated with 50 mM **5** liposomes with or without the addition of Siglec-6 antibody (1:100 dilution) for 2 h and washed with cold PBS before overnight fixation in 2% PFA.

## Lentivirus production
Siglec-6 lentivirus was produced as previously reported by Bhattacherjee, A. et al.[72]. Briefly, $1 \times 10^6$ HEK293T cells were plated in a 6-well dish containing 1.5 mL of DMEM growth medium (Gibco) containing 10% (V/V) fetal bovine serum (FBS; Gibco), 100 U/mL penicillin (Gibco) and 100 μg/mL streptomycin (Gibco). 24 h later, a mixture of 625 ng RP18, 625 ng RP19, 1250 ng hSiglec-6 vector, 7.5 μL TransIT®-LT1 Reagent (Mirus Bio), and Opti-MEM media (Gibco) was added to the HEK293T cells. Cells were incubated with this transfection mixture at 37 °C, 5% CO₂ for 72 h. Following transfection, the cell supernatant was harvested and concentrated using Lenti-X concentrator reagent (Takara Bio) following the manufacturer's instructions.

## Viral transduction
$1.5 \times 10^5$ Daudi cells were plated in a 24-well plate in 250 μL of RPMI growth medium (Gibco) containing 10% (V/V) fetal bovine serum (FBS; Gibco), 100 U/mL penicillin (Gibco) and 100 μg/mL streptomycin (Gibco). A range of 10X concentrated lentivirus (1, 2, 5, 10 μL) was added to the corresponding wells and incubated for approximately 8 h at 37 °C, 5% CO₂ in a tissue culture incubator. After incubation, the media in each well was topped up to 750 μL. Three days post-transduction, 200 μL of cells were plated in a 96-well U-bottom plate and centrifuged at $300 \times g$ for 5 min. The cell pellets were resuspended in 150 μL of flow cytometry buffer (HBSS containing 1% (V/V) FBS, 500 μM EDTA) and the titer of each virus was determined by measuring the mAmetrine+ cells in each well by flow cytometry. The mAmetrine+ cells, ranging from 0.5% to 5%, were re-plated in six-well plates. The mAmetrine+ virally transduced cells were selected for using 300 μg/mL zeocin until the mAmetrine+ population was ≥95%.

## Imaging flow cytometry and quantification

Approximately $2 \times 10^5$ Daudi cells/well were plated into a 96-well U-bottom microplate and centrifuged at $300 \times g$ for 5 min. The cell pellets were placed on ice, and 50 μL of 100 μM of liposome or 1:20 μL of EVs in RPMI growth medium (Gibco) containing 10% (V/V) fetal bovine serum (FBS; Gibco), 100 U/mL penicillin (Gibco), and 100 μg/mL streptomycin (Gibco) was added to their corresponding wells. Plates were incubated for 1 h at either 4 or 37 °C. After incubation, plates were centrifuged at $300 \times g$, 5 min, 4 °C then incubated with fluorescently conjugated anti-Siglec-6 antibody (Alexa Fluor 488; 1:250 dilution; R&D Systems) for 25 min at 4 °C. The plates were centrifuged once more at $300 \times g$, 4 °C, 5 min and the resulting cell pellet was resuspended in 40 μL of flow buffer (1% FBS, 500 μM EDTA Hank's Balanced Salt Solution pH 7.4). 5000 events were collected for each sample using the ImageStream®X Mk II Flow cytometer (excitation lasers 488 nm and 642 nm, ×60 magnification). Data analysis was performed using IDEAS software, version 6.2.

## Placenta explant immunofluorescence imaging

Following fixation, tissue was washed and blocked with 5% normal donkey serum and 0.3% Triton x100 in PBS and incubated with fluorescently conjugated Siglec-6 antibody (Alexa Fluor 594; 1:200 dilution; R&D Systems) overnight at 4 °C. Samples were then washed and incubated with donkey anti-mouse-AF594 (Alexa Fluor® 594; 5 μg/mL; Invitrogen) and fluorescently conjugated phalloidin (iFluor™ 405; 1:400; AAT Bioquest) for 2 h at room temperature and protected from light. Following incubation, explants were washed with 1X PBS-Tween-20 and PBS and mounted on slides with imaging spacers. 1 μm z-stack slices were taken with a Zeiss LSM700 confocal microscope with a Zeiss Plan Apochromat ×63 lens (NA 1.4). Quantification of images was done using Volocity Acquisition Software (Quorum Technologies). For co-localization assays, puncta were defined as objects >0.1 and <1.5 μm. Total number of puncta per μm³ and the number of puncta colocalized with Siglec-6 per μm³ were normalized to liposomal control. For blocking assays, puncta were defined as objects >0.1 and <1.0 μm. One sample t and Wilcoxon test were run to test the significance of the change of normalized total number of puncta per μm³ in the blocking condition compared to non-blocking control. Microscopy Images were processed with ImageJ.

## Human extracellular vesicle (EV) isolation and labeling

Human peripheral blood, taken under an approved institutional ethics protocol, was centrifuged at $1700 \times g$ for 5 min at 4 °C. The upper phase (plasma) was collected and centrifuged at $10,000 \times g$ for 30 min at 4 °C. Following 0.22 mm filtration (Millipore), the supernatant was diluted 10 times using PBS and ultra-centrifuged at $110,000 \times g$ for 2 h at 4 °C using a Type 70 Ti rotor (Beckman Coulter). The pellet containing the EVs was resuspended in PBS, aliquoted, and stored at −80 °C until further use. For EV fluorescent labeling, EVs were incubated with 1.6% (V/V) of NHS-Alexa (Alexa Fluor™ 647 or 488 NHS Ester (Succinimidyl Ester); 10 mg/mL in DMSO stock, Thermo Fisher) or pHrodo (pHrodo™ Red, (Succinimidyl Ester); 10 mg/mL in DMSO stock; Thermo Fisher) overnight at 4 °C or 1 h at room temperature, respectively. Excess dye was removed by spin-filtration, using Ultra-0.5 Centrifugal Filter Units (Millipore). EVs were recovered in PBS, aliquoted, and stored in −80 °C. For EV de-sialylation, 10 μg of EVs were incubated with 5 μL of neuraminidase A or S for 1 h at 37 °C in PBS.

## Culturing of LAD2 cells

The LAD2 cell line (a gift from Arnold Kirshenbaum) was cultured in StemPro-34 SFM media (Life Technologies) supplemented with 2 mM L-glutamine, 100 U/ml penicillin, 50 mg/ml streptomycin, and 100 ng/ml recombinant human SCF (PeproTech, Rocky Hill, NJ). Cells were maintained at $1 \times 10^5$ cells/ml at 37 °C and 5% $CO_2$ and periodically tested for expression of c-KIT and FcεRI by flow cytometry.

## Generation of N2a β1-4GalNT1$^{-/-}$ cells

The N2a β4GalNT1$^{-/-}$ cell line was generated using CRISPR/Cas9. Briefly, guide RNAs were designed to target the β1-4galnt1 gene and mixed with equimolar quantities of Cas9 to construct ribonucleoprotein complexes which were then transfected into N2a cells using Lipofectamine CRISPRMAX (Thermo Fisher, USA) according to manufacturer's instructions. After 24 h of incubation, cells were sorted for the presence of the ATTO 550 fluorescent marker on the tracrRNA, using FACS Aria™ III cell sorter (BD Biosciences, USA), at the Faculty of Medicine and Dentistry Flow Cytometry Facility, University of Alberta, Canada. To generate clonal cell lines, positive cells were sorted one cell per well into a 96-well plate and further expanded. Confirmation of gene knock-out was obtained by immunoblotting and PCR.

## Culturing Neuro2a mouse neuroblastoma cells (N2a) and N2a β1-4GalNT1$^{-/-}$ cells

Cells were cultured in DMEM (Cytvia Life Sciences): Opti-MEM I (1:1) supplemented with 10% heat inactivated fetal bovine serum, 2 mM L-glutamine and 0.11 g/L sodium pyruvate and maintained in a 5% $CO_2$ atmosphere at 37 °C.

## EV isolation from N2a cell lines

N2a and N2a β1-4galnt1$^{-/-}$ cells were incubated in DMEM supplemented with 2 mM L-glutamine and 0.11 g/L sodium pyruvate in the absence of serum for 48 h to prevent uptake of gangliosides from the fetal bovine serum. After 48 h, cells were labeled with the lipophilic dye DiD, as previously described[73]. Briefly, 5 μL of DiD (Thermo Fisher) was added to each mL of cell suspension containing $1 \times 10^6$ cells in DMEM and incubated for 20 min at 37 °C, followed by centrifugation at $400 \times g$ for 5 min at room temperature. The stained cell pellet was further subjected to three rounds of centrifugation in a medium containing serum to remove unbound dye. One 500 cm² dish per cell type was seeded with $14 \times 10^6$ cells each in phenol red-free DMEM:Opti-MEM I (1:1) supplemented with 1X N-2 supplement, 2 mM L-glutamine and 0.11 g/L sodium pyruvate filtered through a 0.1 μm polyethersulfone filter for 24 h.

Cell debris and apoptotic bodies were removed from the conditioned medium by centrifugation at $2000 \times g$ for 10 min at 4 °C in an Eppendorf® Centrifuge 5810R, using an A-4-62 swinging bucket rotor. The EVs remaining in the cleared conditioned medium were isolated by sequential ultrafiltration and size-exclusion chromatography. Briefly, Amicon® Ultra-15 Centrifugal Filters (100,000 MWCO) were used to concentrate the cleared conditioned medium. To minimize EV loss, the filter membranes were first blocked by centrifugation with 5% Tween-80[74] in DPBS at $2600 \times g$ for 10 min at 4 °C, followed by three centrifugations in DPBS for 5 min each. Once blocked, the membrane was kept in DPBS until use to prevent drying. The supernatant containing EVs was concentrated by centrifugation at $2600 \times g$ at 4 °C until the concentrate volume reached 500 μL.

The concentrate was applied to a qEVoriginal Gen 2 Size Exclusion Column and fractions were collected using the Automatic Fraction Collector V1 (iZon Science®). The buffer volume was set to 2.9 mL and thirteen 0.5 mL fractions were eluted with DPBS. The presence of EVs in the fractions was determined by measuring DiD fluorescence ($\lambda_{Ex}/\lambda_{Em} = 644/674$ nm) in each fraction using a SpectraMax® i3x multi-mode microplate reader (Molecular Devices, USA). The fractions enriched with EVs (fractions 1–4) were pooled and concentrated using Amicon® Ultra-4mL Filters (10,000 MWCO) that were blocked with Tween-80 as described above. Purified and concentrated EVs were then stored at −70 °C until use.

## Transmission electron microscopy (TEM)

EVs and liposomes were suspended in PBS or Millipore water, respectively, were placed onto a 300 mesh formvar/copper-coated grid (Ted Pella) and left for 3 min for liposomes and 5 min for EVs.

Then, the excess liquid was removed, and a 10 µl drop of 4% uranyl acetate solution was placed onto the grid and left for 5 and 1 min for liposomes and EVs, respectively. The excess liquid was removed, and the grids were left to dry completely. The grids were analyzed by a Morgagni 268 transmission electron microscope at 80 kV with a Gatan Orius CCD camera.

### Data collection software

Flow cytometry data were collected with BD FACSDivaTM software Version 8.0.1 and analyzed with FlowJo LLC. Version 10.5.3. Xcalibur (Thermo Fisher Scientific, version 4.4) was used for mass spectrometry data acquisitions. ELISA and bead assay data were collected using Molecular Devices Soft Max Pro 7.0.3. Dynamic Light Scattering was performed on Malvern Panalytical Zetasizer software. Microscopy images were processed Velocity software version 6.3 from Quorum technologies.

### Statistical analyses

For datasets comparing only two conditions, a Student's $t$-test was used. When datasets had three or more conditions a Brown–Forsythe and Welch one-way ANOVA was used. All statistical analysis was carried out using GraphPad Prism version 8.4.

### Chemical synthesis

**General.** All reagents were purchased from commercial sources and were used without further purification. THF used in reactions was purified by successive passage through columns of alumina and copper under nitrogen. All reactions were monitored by TLC on silica gel 60-F254 (0.25 mm). Visualization of the reaction components was achieved using UV fluorescence (254 nm) and/or by charring with acidified $p$-anisaldehyde solution in ethanol. Organic solvents were evaporated under reduced pressure below 40 °C, and the products were purified by flash column chromatography on silica gel (230–400 mesh), reverse-phase flash column chromatography (C18), or size exclusion column chromatography (Sephadex-LH-20). HPLC grade $CH_3OH$ was used in the reactions as well as all column purifications. $^1H$ NMR spectra were recorded at 700, 600, or 500 MHz, and chemical shifts were referenced to either TMS (0.0, $CDCl_3$) or $CD_3OD$ (3.30, $CD_3OD$), or HOD (4.78, $D_2O$). $^1H$ data were reported as though they were first order. $^{13}C$ NMR spectra were recorded at 125 MHz and $^{13}C$ chemical shifts were referenced to external acetone (31.07, $D_2O$). Electrospray mass spectra (HRMS-ESI) were recorded on samples suspended in mixtures of THF with $CH_3OH$ and added NaCl.

### Reporting summary

Further information on research design is available in the Nature Portfolio Reporting Summary linked to this article.

## Data availability

The data that support this study are available from the corresponding authors upon request. For access to the raw data please contact the corresponding author (M.S.M.) which is kept electronically and will be forwarded upon request. Source data are provided with this paper.

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

## Acknowledgements

M.S.M. thanks funding from NSERC and GlycoNet for funding, as well as a Canada Research Chair in Chemical Glycoimmunology. E.N.S. thanks Alberta Innovates and NSERC for a student fellowship. L.K.M. and D.L. thank funding through a Canada Excellence Research Chair in Glycomics. T.L.L. thanks NSERC and GlycoNet for funding. S.S. thanks NSERC and GlycoNet for funding. We thank T. Angata (Academia Sinica) for providing the anti-Sig15 antibody. We thank C. Rademacher (University of Vienna) for their feedback and useful discussions. We thank Dr. G. Eskandari-Sedighi and K. Norton (Advanced Microscopy Facility, University of Alberta) for help with microscopy and TEM analyses, respectively.

## Author contributions

E.N.S. and M.S.M. developed the concept and wrote the first draft of the manuscript. D.L. and L.K.M. carried out and interpreted studies with EVs and contributed to binding studies with the placenta. J.J. helped prepare the library of Siglec-expressing CHO cells. M.J. and T.L.L. synthesized the neoglycolipids. X.Y.G. assisted ENS with GL-binding studies. J.N. and M.R.R. carried out GL binding to the placenta. K.A.M. developed the Siglec-6 Daudi cells and performed imaging flow cytometry. J.M. isolated EVs from N2a cells. J.R.E. assisted with the binding studies to human memory B-cells. G.C.D. and F.M. synthesized glycoconjugates used in these studies. E.N.K. and J.S.K. carried out quantitative MS binding studies with Siglec-1. L.H. analyzed the compositions of gangliosides from different sources. A.W. helped in the optimization of the liposomal formulation. A.R.K., C.R.C., and B.P.H. created the β1-4GalNT–/– cells. M.K. provided the LAD2 cells. L.J.W. provided human spleen samples. S.S. provided glycolipid reagents and helped in the early conceptualization of the studies. M.S. and R.D. conducted and interpreted LiGA results. All authors read and contributed to the writing of the manuscript.

## Competing interests

A patent has been filed on this subject with several authors listed as inventors (E.N.S., D.L., M.J., J.N., T.L.L., L.K.M., M.R.R., M.S.M.), and there is potential for future financial benefits to the inventors. All remaining authors declare no competing interests.
