## [Peer Review File · Nature Communications]

Siglec-6 mediates the uptake of extracellular vesicles through a noncanonical glycolipid binding pocketReviewers' Comments:

Reviewer #1:

Remarks to the Author:

In this manuscript, Schmidt et al. evaluate the binding of Siglec family proteins against a panel of glycolipid-derived liposomes and describe the novel identification of Siglec-6 as a selective binder of a synthetic glycolipid (and potentially native glycolipids). As the Siglec family of proteins are actively being exploited for their immunomodulatory properties and the historical difficulties of capturing the interactions between glycolipids and proteins, the study is of significance. An important theme addressed throughout the paper is the observed differences in affinity between Siglecs and ganglioside interactors based on experimental formats (Mahal and Gildersleeve have previously shown that the format of presenting glycans can dramatically affect their densities and three-dimensional presentations, leading to differing interpretations of glycan-binding). The results focused on Siglec-6 as it had not been known to bind gangliosides, and they found that a non-canonical arginine residue was important for binding to a GM3 derivative in a somewhat motif-dependent manner. To show some biological function, the authors show that engagement of Siglec-6 with extracellular vesicles including the synthetic sialoglycolipid results in internalization in transfected Siglec-6 transfected Daudi cells, hinting at possibly new roles of Siglec-6 as an internalization receptor, although in an artificial system. While the biological relevance of the Siglec-6 to glycolipid interaction is not well explored, this work is important – as the fundamental study of binding interactions with glycolipids and proteins is an underexplored research topic. It is difficult/impossible to quantitatively define the dimensionality, density, and presentation of glycolipids presented on the cell membrane, which they have done here. The artificial liposomes presented here are a step above the formats tested. It is clear this is just the first step towards what might be a fruitful research area.

Major points

- It is clear that the glycolipids and the resulting liposomes alone are a significant advancement as a platform to allow estimation of glycolipid-Siglec interactions, but perhaps the Siglec-6 portion requires additional shoring up.
- Perhaps what is most needed for this portion is the reciprocal evidence that native Siglec-6 binds native glycolipids from EVs/cells in a glycan-dependent manner. In the current set of data, one partner (Siglec6 in Fig.6 or the glycolipid in Fig. 5) is presented in an artificial system, where protein expression is over-abundant and the liposomes and the densities they enforce on the synthetic glycolipid are not endogenous. They have the native Siglec-6 expressing placental explant tissue cultures and the isolated EVs that contain the required glycolipid.
- Neoglycolipid 5 is derived from GM3, but GM3 was not identified as a ligand of Siglec-6 in the CBA. Thus, one could infer that the binding motif in 5 does not recapitulate normal binding of Siglec-6 to endogenous ligands. This is further evidenced by the decrease in binding when GM1 was converted to nGLL 4.
- In Fig 6, are differences in WT and mutant EV binding due changes in expression of Siglec-6 mutants or was abundance standardized as performed in S24E? Since mutation resulted in significant reduction of Siglec-6 expression, the relative abundance should be acknowledged if it could be a contributing factor to decreased binding.
- Was the expression level of Siglec-6 considered in the Daudi cells? Is this expression similar to memory-B cells to closely recapitulate physiological relevance. In these experiments the lipid and the binding site are not controlled for. Also, based on the title of the manuscript, there should be a strong connection between the noncanonical binding site and the EV uptake studies. While other experiments do clearly show nGLL binding at R92, in this more physiologically relevant system there could be other binding sites or contributing factors affecting EV internalization. Could the Daudi experiments be repeated with R92 mutants and/or nGLL blockade to say with confidence that EV internalization is mediated by R92.
- Many others including the Varki lab have greatly emphasized the needs for membrane clustering (see Cohen, Varki, et al.) to observe binding. Some further discussion/context on Ten Feizi's work in neoglycolipid membrane arrays could also enhance the story. Have the authors tested this system?

Perhaps it is not the bilayer nature that is required. What about micelles? While the liposomes are formulated to contain defined compositions of glycolipids, the resulting nanoparticles may not have similarly loaded at equal levels by the panel of ganglioside glycans. How do the authors ensure that there is equal loading?

- A model of Siglec-6 displaying interactions (docking/MD) with the lipid portion of the glycolipids would significantly enhance the story.
- A saturation binding curve is expected for a binary equilibrium. The binding is actually not "plateaued." Would zeta potential provide a correlative measure of steric crowding?
- The authors refer to Fig 1c as "bimodal," but there are no two maximal points observed?

Minor points:

- In Fig 3D, E one cannot distinguish if the phosphatidylethanolamine linker or the acyl chain abrogates Siglec-6 binding as the ether linkage is both motifs are changed. These experiments do clearly show that the sialic acid linkage is important, which later reinforces the data in Fig 6E, but this panel here is not broad enough to comment on acyl chain and linker separately.
- In Fig. 5, the pink and purple color schemes are not very friendly to color-impaired readers. In 5f the number of colocalized liposomes is nearly double the number of total liposomes in Fig. 5e. Can this discrepancy be explained?
- The number of abbreviations in this manuscript (CBA, BBA, LIGA, UT) is somewhat excessive and distracting.
- The re-coining of many standard assays (LOLA, BBA, CBA) is also a bit cumbersome, some re-consideration is warranted.
- The same group has previously published and described the mutant CHO cell lines that express Siglec6 (ref. 30). As such, it is no longer "a novel panel."
- Including both the bar graph and histogram representation of liposome binding is a bit jarring – suggest to include either one in the main text and relegate the other representation in the Supporting Information.
- Has the specificity of the Anti-Siglec-6 antibody been evaluated?

Reviewer #2:

Remarks to the Author:

Understanding the glycolipid ligand preferences of siglecs is of great interest, since these sialoside-binding proteins play important roles in modulating immune cell activation. Indeed, they represent promising targets in both cancer and autoimmune disorders.

In the past, these ligand preferences have been explored using recombinant protein and surface-immobilized glycolipids. In this work, Macauley and co-workers optimize the use of glycolipid liposomes and siglec-expressing CHO cells to map the sialoglycolipid-binding properties of human siglecs with both ligand and receptor presented in a physiologically relevant sense.

This approach enabled the discovery of a hitherto unknown auxiliary sialoglycolipid binding site on siglec-6. The authors demonstrate that this interaction mediates recognition of extracellular vehicles and can be exploited to deliver liposomes to siglec-6 expressing primary cells. The latter finding could be particularly valuable in future efforts to manipulate memory B cells in various disease settings.

Overall, the experimental work is of excellent quality and the findings of sufficient significance to the field to merit publication in Nature Communications. Nonetheless, I have a few comments that I think the authors should consider addressing in order to improve the manuscript.

Comments:

The following sentence may need to be explained better – it's conclusion was not self-evident to me:
"The divergent behavior of GM3 from the other gangliosides suggested that the binding between Siglec-1 and the GM3 trisaccharide is intrinsically weaker than the oligosaccharides of the other three gangliosides that were examined"

In Figure 2a, some cells are 'demasked' by neuraminidase treatment but others are not. In an ideal world the protocol would be the same for all points in the heat map so they can be directly compared, although I appreciate that would be a lot of work. However, I do think it is necessary for the RA mutant controls for siglec-4/5/7/9, since absence of any binding in these samples could be due to the same cis-ligand binding that masked the activity of the wild-type receptors.

In the following sentence 'conserved' should be 'conservative':

"As the R92A mutant showed minimal expression, a more conserved R92K mutant was used and gave cell surface expression levels at 42% compared to WT Siglec-6"

It would be good to make the R92K siglec6-Fc construct to confirm the conclusions in the following sentences using the LOLA and BBA protocols outlines in Fig 2.

"To be certain, we performed careful gating on equivalent levels of Siglec-6 expression between WT and R92K Siglec-6 and confirmed that R92K Siglec-6 shows minimal interactions with 5 nGLLs (Supplementary Fig.24e). These results demonstrate that glycolipid binding is not dependent on the canonical Arg122 and that Arg92 is the critical residue mediating glycolipid recognition."

Reviewer #3:

Remarks to the Author:

This study presents the highly interesting observation that the immune-regulatory receptor Siglec-6 recognizes sialylated glycolipids in liposomes and possibly extracellular vesicles through a non-canonical binding site. Siglecs are generally thought to bind sialic acid capped glycans through the canonical binding involving a conserved Arg (Arg122 in Siglec-6), and the study convincingly demonstrates that Arg122 is not required for the observed ganglioside binding and rather a separate binding site including Arg92 is involved. The study is truly a scholarly contribution with elaborate and careful examination of the presentation and recognition of glycolipids in liposomes using multiple assay formats, and the results presented from these are widely useful for the field. New resources are presented including CHO cells expressing Siglecs and neoglycolipids. The manuscript is clear and well written, and the conclusions drawn fully supported. The authors decided (clearly for good reasons) to limit the report to Siglec-6 and its non-canonical binding site and glycolipids, but the screen of all Siglecs provides "food" for further thoughts and discussion.

Specific suggestions:

The abstract should be improved. E.g. "...interactions, we discovered many new interactions, most notably Siglec-6." Last part is not interpretable? "A tailored ability of Siglec-6 to recognize glycolipids in a bilayer is enabled by a noncanonical binding site." Should probably be presented as a hypothesis?

The results presented in Figure 2A for all Siglecs could be discussed further despite the focus on Siglec-6? Do the glycolipid binding profiles correlate with previous reports and is it possible to draw conclusions on binding to glycan epitopes on different types of glycoproteins?

The authors should discuss potential roles of the canonical binding site of Siglec-6? In the Discussion the sentence "It was previously reported that the canonical binding site in Siglec-6 recognizes α -(2 \rightarrow 6)-linked sialosides⁴²; however, our results from the nGLL binding assays as well as the neuraminidase S treatment of EVs suggest that this noncanonical binding site in Siglec-6 prefers α -(2 \rightarrow 3) sialosides." Is unclear. Do we really know if the canonical binding site does not bind e.g. 2-6?

The authors should also discuss potential glycoprotein ligands for Siglec-6?

In the Figures it may be helpful for the readers if the sialic acids are shown with the correct linkages (like in Fig. 3). At least its confusing that only GM3 and GD3 is shown as linear structures?

Reviewer #1 (Remarks to the Author):

In this manuscript, Schmidt et al. evaluate the binding of Siglec family proteins against a panel of glycolipid-derived liposomes and describe the novel identification of Siglec-6 as a selective binder of a synthetic glycolipid (and potentially native glycolipids). As the Siglec family of proteins are actively being exploited for their immunomodulatory properties and the historical difficulties of capturing the interactions between glycolipids and proteins, the study is of significance. An important theme addressed throughout the paper is the observed differences in affinity between Siglecs and ganglioside interactors based on experimental formats (Mahal and Gildersleeve have previously shown that the format of presenting glycans can dramatically affect their densities and three-dimensional presentations, leading to differing interpretations of glycan-binding). The results focused on Siglec-6 as it had not been known to bind gangliosides, and they found that a non-canonical arginine residue was important for binding to a GM3 derivative in a somewhat motif-dependent manner. To show some biological function, the authors show that engagement of Siglec-6 with extracellular vesicles including the synthetic sialoglycolipid results in internalization in transfected Siglec-6 transfected Daudi cells, hinting at possibly new roles of Siglec-6 as an internalization receptor, although in an artificial system. While the biological relevance of the Siglec-6 to glycolipid interaction is not well explored, this work is important – as the fundamental study of binding interactions with glycolipids and proteins is an underexplored research topic. It is difficult/impossible to quantitatively define the dimensionality, density, and presentation of glycolipids presented on the cell membrane, which they have done here. The artificial liposomes presented here are a step above the formats tested. It is clear this is just the first step towards what might be a fruitful research area.

Reply: Thank you for the positive assessment of our work.

Major points

1) It is clear that the glycolipids and the resulting liposomes alone are a significant advancement as a platform to allow estimation of glycolipid-Siglec interactions, but perhaps the Siglec-6 portion requires additional shoring up.

Reply: Thank you for recognizing the novelty of our work. We thank the reviewer for the helpful suggestions below that stimulated further experiments to strengthen the role of Siglec-6 in binding glycolipids.

2) Perhaps what is most needed for this portion is the reciprocal evidence that native Siglec-6 binds native glycolipids from EVs/cells in a glycan-dependent manner. In the current set of data, one partner (Siglec6 in Fig.6 or the glycolipid in Fig. 5) is presented in an artificial system, where protein expression is over-abundant and the liposomes and the densities they enforce on the synthetic glycolipid are not endogenous. They have the native Siglec-6 expressing placental explant tissue cultures and the isolated EVs that contain the required glycolipid.

Reply: This is an excellent point. We have performed several additional experiments to address this.

Experiment 1: We isolated EVs from WT and β 1-4GalNT1^{-/-} cells, the latter of which do not make complex gangliosides. These new results are presented in Fig. 6g of our revised manuscript and demonstrate that EVs from β 1-4GalNT1^{-/-} cells bind less well to Siglec-6. In the text of our modified manuscript, we summarize these results as follows:

“To examine whether gangliosides in EVs mediate binding to Siglec-6, we prepared EVs from WT and β 1-4GalNT1^{-/-} N2a cells, as these knockout cells cannot synthesize complex gangliosides, which are the ligands for Siglec-6 (Supplementary Fig. 41). A 68% reduction in binding of β 1-4GalNT1^{-/-}-derived EVs compared to WT EVs (Fig. 6g), suggesting that complex glycolipids in EVs support binding to Siglec-6.”

Experiment 2: We demonstrated that 5 nGLLs block the interaction between EVs and Siglec-6. These results are shown in Fig. 6d of our revised manuscript. These results provide further evidence that the newly discovered glycolipid binding site mediates binding to EVs. We summarize these results in our revised manuscript as follows:

“These results suggested that EVs bind to Siglec-6 in the same manner as GLLs. In support of this, 5 nGLLs competed away binding of EVs to Siglec-6 (Fig. 6d, Supplementary Fig. 38).”

Experiment 3: To demonstrate EV binding to Siglec-6 under physiological conditions, we have added binding experiments using LAD2 cells (a mast cell line) in our revised manuscript. First, we compared the Siglec-6 expression between human mast cells and LAD2 cells and found that they were comparable (Fig. 5a). We then demonstrated that LAD2 cells could be targeted via Siglec-6 with 5 nGLLs (Fig 5b). Lastly, in Fig. 6e, we demonstrate that EV binding to LAD2 cells is decreased by the anti-Siglec-6 antibody. In the text of our modified manuscript, we summarize these results as follows:

“A modest, but significant, reduction in EV binding with EVs from two different donors was observed to LAD2 cells blocked with an anti-Siglec-6 antibody (Fig. 6e, Supplementary Fig. 39).”

3) Neoglycolipid 5 is derived from GM3, but GM3 was not identified as a ligand of Siglec-6 in the CBA. Thus, one could infer that the binding motif in 5 does not recapitulate normal binding of Siglec-6 to endogenous ligands. This is further evidenced by the decrease in binding when GM1 was converted to nGLL 4.

Reply: Two pieces of evidence suggest to us that this is not the case:

i) We demonstrate that GM3 in a membrane is a poor ligand for Siglec-6, as well as several other Siglecs. Previous molecular modeling of GM3 in a membrane demonstrates that the trisaccharide of GM3 lays down on the membrane, burying its trisaccharide in a way that may prevent engagement of the ganglioside by the Siglec (Ref 58 of our revised manuscript; DeMarco ML and

Woods RJ, *Glycobiology*, 2009). The triazole linker in **5** nGLL may prevent the burying behavior, forming the basis for why it can serve as a good ligand.

ii) Results from the Siglec-6/8 chimera studies (Supplementary Fig. 23a,c) demonstrated that **5** nGLLs showed the same binding pattern to the chimeras as ganglioside GD1a.

From the data in our first draft of the manuscript, we could not fully rule out the scenario proposed by the reviewer whereby gangliosides (like GM1) and **5** bind to a different site on Siglec-6, therefore, we have carried out one additional experiment:

Experiment 4: GM1 liposomes were shown to competitively inhibit the binding of **5** nGLL Siglec-6, which is presented in Supplementary Fig. **20** of our revised manuscript. We summarize these findings in the text as follows:

“To understand how **5** nGLLs engage Siglec-6, we performed a competition assay between GM1 GLLs and **5** nGLLs. The binding of GM1 GLLs decreased as the concentration of **5** nGLLs increased, suggesting that the two ligands compete for the same binding pocket (Supplementary Fig. 20).”

4) In Fig 6, are differences in WT and mutant EV binding due to changes in expression of Siglec-6 mutants or was abundance standardized as performed in S24E? Since mutation resulted in significant reduction of Siglec-6 expression, the relative abundance should be acknowledged if it could be a contributing factor to decreased binding.

Reply: Our original data did not standardize based on differences in expression. We thank the reviewer for pointing this out. In the revised manuscript, standardized expression is now shown in (Fig. 6b, Supplementary Fig. 36a,b).

5.1) Was the expression level of Siglec-6 considered in the Daudi cells? Is this expression similar to memory-B cells to closely recapitulate physiological relevance.

Reply: The reviewer is correct that Siglec-6 levels on the transduced Daudi cells are higher than *bona fide* memory B cells. Therefore, we have mentioned that this represents overexpressed conditions in the text of our revised manuscript:

“...we investigated if Siglec-6 can internalize cargo such as EVs. Using Daudi cells, a human B-cell line, transduced to overexpress Siglec-6...”

To target Siglec-6 under more physiological expression levels, we carried out the experiments on LAD2 cells, as described above, in ***Experiment 3***. Siglec-6 expression levels in LAD2 cells are also compared to Siglec-6 expression cells on primary splenic mast cells. This experiment demonstrates that, if anything, LAD2 cells express slightly less Siglec-6 than primary mast cells.

5.2) *In these experiments the lipid and the binding site are not controlled for. Also, based on the title of the manuscript, there should be a strong connection between the noncanonical binding site and the EV uptake studies. While other experiments do clearly show nGLL binding at R92, in this more physiologically relevant system there could be other binding sites or contributing factors affecting EV internalization. Could the Daudi experiments be repeated with R92 mutants and/or nGLL blockade to say with confidence that EV internalization is mediated by R92.*

Reply: This comment is fundamentally related to the reviewer's second comment. As suggested by the reviewer, we carried out **Experiment 2**, in which 5 nGLL was tested for its ability to compete for binding to Siglec-6 with EVs. The results in Fig. 6d demonstrate that 5 nGLL and EVs clearly compete. We also carried out two other experiments.

Experiment 5: The R92K Siglec-6 mutant was expressed as an Fc chimera, enabling equal loading on beads compared to WT Siglec-6. This experiment is presented in Fig. 6c of our revised manuscript and shows that R92K Siglec-6 cannot bind glycolipids.

Experiment 6: The Daudi cell internalization experiments were repeated with R122A Siglec-6 (Fig. 6h) and no significant difference was observed between this mutant and WT Siglec-6, strongly suggesting that EVs are not internalized via the canonical binding site.

6.1) *Many others including the Varki lab have greatly emphasized the needs for membrane clustering (see Cohen, Varki, et al.) to observe binding. Some further discussion/context on Ten Feizi's work in neoglycolipid membrane arrays could also enhance the story. Have the authors tested this system?*

Reply: We have not tested Ten Feizi's system, but it is a good suggestion that we will consider as a future direction. We have expanded the discussion by adding the following:

"The effect of glycan density on Siglec binding has been well established⁶⁰ and in line with this, all the approaches used in this work, with the exception of the mass spectrometry based Siglec binding assay, leverage avidity."

6.2) *Perhaps it is not the bilayer nature that is required. What about micelles?*

Reply: We performed an experiment with GM1 micelles to address this point:

Experiment 7: We used GM1 at a concentration higher than its critical micellar concentration and tested whether it could disrupt binding of GM1 liposomes to Siglec-1 in the cell assay. These results are presented in Supplementary Fig. 4 of our revised manuscript and demonstrated that micellular GM1 does not effectively compete against GM1 liposomes. There are two likely possible explanations: (i) micelles are not a true lipid bilayer and, therefore, presentation may not be optimal; (ii) GM1 micelles are composed entirely of GM1 and, therefore, the steric crowding observed with high amounts of GM1 in the liposomes may also be at play. We summarize these results in our revised manuscript as follows:

“Ganglioside micelles (GM1) showed no binding to Siglec-1 (Supplementary Fig. 4).”

6.3) *While the liposomes are formulated to contain defined compositions of glycolipids, the resulting nanoparticles may not have similarly loaded at equal levels by the panel of ganglioside glycans. How do the authors ensure that there is equal loading?*

Reply: This a very good suggestion, which we have investigated.

Experiment 8: The incorporation of GM1 into liposomes has been carefully investigated by Han *et al.* (ref 40) using deuterated GM1. Using their approach, we also quantified in incorporation efficiency of GM2 and GM3 and found that GM2 and GM3 are incorporated into liposomes at 98% and 96% respectively. These results are summarized in Supplementary Table 1.

7) *A model of Siglec-6 displaying interactions (docking/MD) with the lipid portion of the glycolipids would significantly enhance the story.*

Reply: We agree that this would be a great addition. We have initiated a collaboration with Prof. Elisa Fadda to do this just this. However, this modelling is computationally intense and will require a significant effort and time. Therefore, we feel that modeling is beyond the scope of this study but will make for an excellent follow-up investigation.

8) *A saturation binding curve is expected for a binary equilibrium. The binding is actually not “plateaued.” Would zeta potential provide a correlative measure of steric crowding?*

Reply: We agree that plateau is not the best descriptor; therefore, we have changed “plateaued” to “was maximal at”, throughout. To the best of our knowledge, zeta potential cannot be used to study ganglioside clustering in a membrane. We plan to carry out more detailed biophysical measurements, with the help of a collaborator, in the future to investigate this phenomenon.

9) *The authors refer to Fig 1c as “bimodal,” but there are no two maximal points observed?*

Reply: Thank you for pointing out this incorrect word usage. We have changed: ‘bimodal’ to ‘unimodal’ throughout.

Minor points:

1) *In Fig 3D, E one cannot distinguish if the phosphatidylethanolamine linker or the acyl chain abrogates Siglec-6 binding as the ether linkage is both motifs are changed. These experiments do clearly show that the sialic acid linkage is important, which later reinforces the data in Fig 6E, but this panel here is not broad enough to comment on acyl chain and linker separately.*

Reply: The point is well taken. We have added the following statement to the results section:

“The presentation of the oligosaccharide with respect to the bilayer may explain why genuine GM3 GLLs did not engage Siglec-6 whereas 5 nGLLs did. We speculate that the combination of the triazole-linkage and the di-O-hexadecyl glycerol lipid anchor more optimally presents the trisaccharide from the bilayer for engagement by Siglecs compared to genuine GM3, which is likely buried in the bilayer⁵⁸. However, the scope of our panel was not wide enough to resolve the contribution of each component to the presentation of the oligosaccharide.”

2.1) *In Fig. 5, the pink and purple color schemes are not very friendly to color-impaired readers.*

Reply: Thank you for pointing this out. The color scheme has been adjusted.

2.2) *In 5f the number of colocalized liposomes is nearly double the number of total liposomes in Fig. 5e. Can this discrepancy be explained?*

Reply: Results in Fig. 5f and g (formally Fig. 5e and f) were carried out using a different set of placentas than Fig. 5f; therefore, differences likely represent small differences between biological replicates. As each point represents tissue from separate individuals, the large difference is driven by one sample that had high numbers. We performed statistical analyses to determine if that point was a statistical outlier, but it did not reach statistical significance and was, therefore, included. We have added a sentence to the figure legend to help the reader understand that these points:

“For panels f, g, and h, each datum is representative of a different donor, which may introduce variability between panels.”

3) *The number of abbreviations in this manuscript (CBA, BBA, LIGA, UT) is somewhat excessive and distracting. The re-coining of many standard assays (LOLA, BBA, CBA) is also a bit cumbersome, some re-consideration is warranted.*

Reply: We understand the concern of the reviewer. We used these to frequently compare and contrast between results obtained in the different assays. To alleviate the concerns of the reviewer, we have renamed ‘CBA’ to ‘cell assay’ and ‘BBA’ to ‘bead assay’.

4) *The same group has previously published and described the mutant CHO cell lines that express Siglec6 (ref. 30). As such, it is no longer “a novel panel.”*

Reply: The cell lines in Ref. 30 (Rodrigues *et al.*, *Nat Comm*, 2020) are for soluble versions of the Siglecs as Fc chimeras. These new CHO cell lines are the full-length versions of the Siglecs, which were newly created by our group specifically for this work. We have also adjusted the text to try to clarify this point.

“A novel panel of 24 Chinese Hamster Ovary (CHO) cells were developed where each cell line expresses a full-length, membrane-bound wildtype (WT) human Siglec...”

5) Including both the bar graph and histogram representation of liposome binding is a bit jarring – suggest to include either one in the main text and relegate the other representation in the Supporting Information.

Reply: We have removed the histograms from most panels in Fig. 6. We would like to keep the representative histograms in Fig. 1b and Fig. 6g as it shows the reader what the raw data looks like. Our rationale for showing both the raw data and histograms in the Supplemental Figures is to show both the raw data and quantification. It is our preference to keep these two presentation formats for the key main text figures where we have included both.

6) Has the specificity of the Anti-Siglec-6 antibody been evaluated?

Reply: In our hands, the antibody does not recognize untransfected cells (CHO or Daudi). Moreover, the antibody was able to reduce 5 nGLL binding CHO cells, memory B-cells, LAD2 cells and placenta. We have not observed any indication that it binds another target. Moreover, in our experiments on primary human white blood cells where most subtypes of white blood cells were identified, we did not observe Siglec-6 staining to any cell types other than cells that have been previously identified as Siglec-6 expressing cells (memory B-cells and mast) suggesting that the anti-Siglec-6 antibody is specific for Siglec-6.

Reviewer #2 (Remarks to the Author)

Understanding the glycolipid ligand preferences of siglecs is of great interest, since these sialoside-binding proteins play important roles in modulating immune cell activation. Indeed, they represent promising targets in both cancer and autoimmune disorders. In the past, these ligand preferences have been explored using recombinant protein and surface-immobilized glycolipids. In this work, Macauley and co-workers optimize the use of glycolipid liposomes and siglec-expressing CHO cells to map the sialoglycolipid-binding properties of human siglecs with both ligand and receptor presented in a physiologically relevant sense. This approach enabled the discovery of a hitherto unknown auxiliary sialoglycolipid binding site on siglec-6. The authors demonstrate that this interaction mediates recognition of extracellular vehicles and can be exploited to deliver liposomes to siglec-6 expressing primary cells. The latter finding could be particularly valuable in future efforts to manipulate memory B cells in various disease settings. Overall, the experimental work is of excellent quality and the findings of sufficient significance to the field to merit publication in Nature Communications. Nonetheless, I have a few comments that I think the authors should consider addressing in order to improve the manuscript.

Reply: We thank the reviewer for recognizing the significance of our work.

Comments:

1. The following sentence may need to be explained better – it's conclusion was not self-evident to me: "The divergent behavior of GM3 from the other gangliosides suggested that the binding

between Siglec-1 and the GM3 trisaccharide is intrinsically weaker than the oligosaccharides of the other three gangliosides that were examined”

Reply: We have revised the text to make this sentence easier to understand.

“One explanation for the weaker binding of GM3 liposomes, compared to other GLLs, is that the intrinsic affinity of Siglec-1 for the GM3 trisaccharide is weaker than the oligosaccharide portion of the other gangliosides.”

2. In Figure 2a, some cells are ‘demasked’ by neuraminidase treatment but others are not. In an ideal world the protocol would be the same for all points in the heat map so they can be directly compared, although I appreciate that would be a lot of work. However, I do think it is necessary for the R→A mutant controls for siglec-4/5/7/9, since absence of any binding in these samples could be due to the same cis-ligand binding that masked the activity of the wild-type receptors.

Reply: We agree that a complete set of masked and unmasked conditions would be ideal, although practically this is an enormous amount of work. To address this and be certain that unmasking of several key Siglecs was not missed, we have carried out unmasking studies with Siglec-4R, 5R, and 9R. No binding of any GLLs was observed to any of these Siglecs after neuraminidase treatment. These new results are now included Fig 2a and in Supplementary Fig. 13.

3. In the following sentence ‘conserved’ should be ‘conservative’: “As the R92A mutant showed minimal expression, a more conserved R92K mutant was used and gave cell surface expression levels at 42% compared to WT Siglec-6”

Reply: Thank you for pointing out this error. The corresponding change has been made.

4. It would be good to make the R92K siglec6-Fc construct to confirm the conclusions in the following sentences using the LOLA and BBA protocols outlines in Fig 2. “To be certain, we performed careful gating on equivalent levels of Siglec-6 expression between WT and R92K Siglec-6 and confirmed that R92K Siglec-6 shows minimal interactions with 5 nGLLs (Supplementary Fig.24e). These results demonstrate that glycolipid binding is not dependent on the canonical Arg122 and that Arg92 is the critical residue mediating glycolipid recognition.”

Reply: This is an excellent suggestion. We have carried out the requested experiment with the Fc chimera of R92K Siglec-6 and results are described in detail above (**Experiment #5**), in response to *Reviewer #1*. In addition, we have gone beyond this to generate more mutants around R92 in an effort to find mutants that do not have perturbed expression levels yet do show a significant reduction in GL binding. These results of this expanded mutagenesis are described in our revised manuscript as follows:

“An Fc chimera of R92K Siglec-6 was also made and used in the bead assay, which demonstrated less than 5% of the binding to 5 nGLLs compared to WT Siglec-6 (Supplementary Fig. 27).

Contributions from amino acids surrounding Arg92 were also investigated and F93A, L95A, and G175M mutants showed significantly decreased binding to 5 nGLLs (Fig. 5e, Supplementary Fig. 28).

Reviewer #3 (Remarks to the Author):

This study presents the highly interesting observation that the immune-regulatory receptor Siglec-6 recognizes sialylated glycolipids in liposomes and possibly extracellular vesicles through a non-canonical binding site. Siglecs are generally thought to bind sialic acid capped glycans through the canonical binding involving a conserved Arg (Arg122 in Siglec-6), and the study convincingly demonstrates that Arg122 is not required for the observed ganglioside binding and rather a separate binding site including Arg92 is involved. The study is truly a scholarly contribution with elaborate and careful examination of the presentation and recognition of glycolipids in liposomes using multiple assay formats, and the results presented from these are widely useful for the field. New resources are presented including CHO cells expressing Siglecs and neoglycolipids. The manuscript is clear and well written, and the conclusions drawn fully supported. The authors decided (clearly for good reasons) to limit the report to Siglec-6 and its non-canonical binding site and glycolipids, but the screen of all Siglecs provides “food” for further thoughts and discussion.

Reply: We thank the reviewer for their positive comments and positive appraisal of our work.

Specific suggestions:

1. The abstract should be improved. E.g. “...interactions, we discovered many new interactions, most notably Siglec-6.” Last part is not interpretable? “A tailored ability of Siglec-6 to recognize glycolipids in a bilayer is enabled by a noncanonical binding site.” Should probably be presented as a hypothesis?

Reply: Thank you for pointing out the ambiguity in this sentence. In our revised manuscript, we have changed it to the following:

“Through optimizing a liposomal formulation to dissect Siglec–glycolipid interactions, it was discovered that Siglec-6 can recognize glycolipids independent of its canonical binding pocket suggesting that Siglec-6 possesses a secondary binding pocket tailored for recognizing glycolipids in a bilayer.”

2. The results presented in Figure 2A for all Siglecs could be discussed further despite the focus on Siglec-6? Do the glycolipid binding profiles correlate with previous reports and is it possible to draw conclusions on binding to glycan epitopes on different types of glycoproteins?

Reply: We agree with the reviewer that there is much that could be discussed about interactions between Siglecs, and glycolipids based on our data in Figure 2A. In our original manuscript, we did so briefly within the discussion so as to keep as much of the focus on Siglec-6: “Ganglioside

interactions with Siglec-1, -4, -5, -7, -9, and -10, were observed in this work and have been previously observed in other studies^{10, 12, 18}.” In our revised manuscript, we have added additional text following this sentence to describe a few specific examples to describe this in more detail:

“Using the ELISA, LOLA, cell, bead, and assays, many of the established Siglec-ganglioside interactions were reproduced, specifically with Siglec-1, -4, -5, -7, -9, and -10^{10, 12, 18}. In addition, we found novel interactions including: Siglec-4 with GM1, GD1b, and GD3; Siglec-5 with GM1; Siglec-7 recognizes GM4; Siglec-9 recognizes GM1; and Siglec-10 recognizes GM1, GM2, and GD3. However, not all experimental platforms revealed the same interactions, and it is important to consider how membrane dynamics influence avidity.”

3. The authors should discuss potential roles of the canonical binding site of Siglec-6? In the Discussion the sentence “It was previously reported that the canonical binding site in Siglec-6 recognizes α -(2→6)-linked sialosides⁴²; however, our results from the nGLL binding assays as well as the neuraminidase S treatment of EVs suggest that this noncanonical binding site in Siglec-6 prefers α -(2→3) sialosides.” Is unclear. Do we really know if the canonical binding site does not bind e.g. 2-6? The authors should also discuss potential glycoprotein ligands for Siglec-6?

Reply: We completely agree with the reviewer. There is much to be learned about whether the canonical binding of Siglec-6 supports sialoside recognition. Some efforts in our lab to uncover canonical binding have not been successful and discussions with other experts in the field suggest that we are not the only ones. We prefer to keep the focus of our manuscript on the non-canonical binding and avoid going in depth into whether or not the canonical site supports glycan recognition. Nevertheless, as the reviewer correctly points out, our results demonstrate that no binding could be detected to several α 2→6 sialosides.

4. In the Figures it may be helpful for the readers if the sialic acids are shown with the correct linkages (like in Fig. 3). At least its confusing that only GM3 and GD3 is shown as linear structures?

Reply: We agree with the reviewer and have re-drawn the symbol nomenclature of the oligosaccharides for each ganglioside to match how they are presented in Fig. 3.

Reviewers' Comments:

Reviewer #1:

Remarks to the Author:

The authors are commended for their outstanding responses to this Reviewer's suggestions. The manuscript as it stands is now suitable for publication in Nature Communications. It is a remarkable piece of study that will significantly advance the study of Siglec-glycolipid interactions, and it will be a much cited resource for future investigations.

Reviewer #2:

Remarks to the Author:

The authors have addressed all of my prior concerns and I think the work is suitable for publication in Nat. Commun.

Reviewer #3:

Remarks to the Author:

All concerns addressed satisfactorily. This is an important contribution to the Siglec field.